**Article**                                                                 https://doi.org/10.1038/s41467-024-47659-w

# NMR characterization of RNA binding property of the DEAD-box RNA helicase DDX3X and its implications for helicase activity

**Yuki Toyama** [1] ✉ **& Ichio Shimada** [1,2] ✉

The DEAD-box RNA helicase (DDX) plays a central role in many aspects of RNA metabolism by remodeling the defined structure of RNA molecules. While a number of structural studies have revealed the atomistic details of the interaction between DDX and RNA ligands, the molecular mechanism of how this molecule unwinds a structured RNA into an unstructured single-stranded RNA (ssRNA) has largely remained elusive. This is due to challenges in structurally characterizing the unwinding intermediate state and the lack of thermodynamic details underlying this process. In this study, we use solution nuclear magnetic resonance (NMR) spectroscopy to characterize the interaction of human DDX3X, a member of the DDX family, with various RNA ligands. Our results show that the inherent binding affinity of DDX3X for ssRNA is significantly higher than that for structured RNA elements. This preferential binding, accompanied by the formation of a domain-closed conformation in complex with ssRNA, effectively stabilizes the denatured ssRNA state and thus underlies the unwinding activity of DDX3X. Our results provide a thermodynamic and structural basis for the DDX function, whereby DDX can recognize and remodel a distinct set of structured RNAs to participate in a wide range of physiological processes.

DEAD-box RNA helicase (DDX) constitutes the largest family of RNA helicases in humans and plays a central role in regulating cellular RNA metabolism, including transcription, processing, transport, and degradation[1,2]. Since many of these processes involve the formation of well-defined RNA structures, DDX regulates these processes by facilitating conformational rearrangements of structured RNAs through binding and hydrolyzing adenosine triphosphate (ATP). As the name suggests, DDX is characterized by the presence of a conserved sequence motif, Asp-Glu-Ala-Asp (DEAD), which plays pivotal roles in ATP binding and hydrolysis. The core function of DDX is typically described as a helicase responsible for unwinding short double-stranded RNA (dsRNA) into single-stranded RNA (ssRNA). However, DDX proteins are involved in a remarkable range of biological processes by binding to diverse RNA substrates, such as self-splicing ribozymes, messenger RNA (mRNA), ribosomal RNA, and micro RNA, thus remodeling the ternary structure of these RNAs and/or translocating these RNA substrates[1,2]. Malfunctions of DDX proteins are frequently associated with various pathogenic processes in humans. One prominent example is DDX3X, a member of the DDX family encoded on the X chromosome. DDX3X primarily participates in translational initiation by remodeling the higher-order structure of the 5′-untranslated region of mRNA[3–6]. DDX3X also plays significant roles in

[1]RIKEN Center for Biosystems Dynamics Research (BDR), 1-7-22 Suehiro-cho, Tsurumi-ku, Yokohama, Kanagawa 230-0045, Japan. [2]Graduate School of Integrated Sciences for Life, Hiroshima University, 1-4-4 Kagamiyama, Higashi-Hiroshima, Hiroshima 739-8528, Japan. ✉e-mail: yuki.toyama@riken.jp; ichio.shimada@riken.jp

regulating inflammatory responses, mRNA transport, and the formation of RNA granules to regulate RNA sub-cellular localization[3,7,8]. As anticipated from the critical roles of DDX3X, mutations in DDX3X are linked to pathogenic processes such as tumor progression in medulloblastoma[7,9–11]. Further, dysfunctions of DDX3X are associated with developmental disorders and intellectual disability[6,12,13].

Given the physiological importance of DDX proteins, it has been of considerable interest to understand how these molecules remodel RNA structures at the molecular level. This has mainly been achieved through solving high-resolution atomistic structures of DDX under various conditions[14–21]. The basic architecture of the DDX core region comprises two well-conserved recombinase A (RecA)-like domains, D1 and D2, which are responsible for the helicase activity and binding to ATP[1] (Fig. 1a). Outside the conserved folded core, each DDX subfamily

contains unique N- and C-terminal regions to exert subfamily-specific roles. Typically, these regions are involved in binding to specific RNA segments and/or participating in protein-protein interactions[1]. A number of structures of the functional core of the DDX proteins, in the presence or absence of RNA and ADP/ATP analogs, have been solved, and the functional cycle of the RNA unwinding process has been proposed by comparing these structures[14–21]. The proposed cycle mainly comprises three distinct states: the RNA-free state, the dsRNA-bound state (pre-unwound state), and the ssRNA-bound state (post-unwound state). In the RNA-free state, DDX adopts an open conformation where D1 and D2 domains tumble independently[14,16,21,22], priming if for binding to RNA substrates. In the post-unwound ssRNA-bound state, the DDX protein adopts a well-defined closed conformation where the substrate RNA simultaneously binds to both domains in a bipartite manner, and this closed configuration is highly conserved among the DDX subfamilies[15,21,23–25]. Compared to the ssRNA-bound state, relatively less is known about the pre-unwound, dsRNA-bound state of DDX. The crystal structure of the truncated D2 domain of yeast DDX, Mss116p, bound to a 14-bp dsRNA was first solved[18], and later, the crystal structure of the complex of the nucleotide-free DDX3X core bound to a 24-bp dsRNA was determined[20]; in both cases, DDX binds to intact dsRNA in the A-form geometry. Notably, characterizations of the pre-unwound state were mostly limited to complexes with dsRNA, even though DDX is involved in the recognition of a diverse set of structured RNAs[1,2]. Recently, Wurm has reported the crystal structure of *E.coli* DbpA in complex with a hairpin RNA substrate[25]. In this structure, DbpA recognizes the 5'-extended ssRNA region, closely resembling the ssRNA-bound, post-unwound state structure.

While these high-resolution structures have provided important details of the intermolecular interactions between DDX and RNA, a number of questions have remained unanswered: how does DDX recognize a diverse set of RNA structures for remodeling, and what is the thermodynamic driving force that facilitates the unwinding of structured RNA elements? The problem lies in the fact that these static structures in essence represent energetically stable complexes, which do not provide the structural details about the high-energy intermediate that involves the transient base opening of structured RNA molecules. Also, quantitative assessments of binding affinity for each RNA structural sub-state are often lacking in previous structural studies, thus limiting our understanding of the thermodynamic basis upon which stably formed dsRNA or structured elements of RNA are transformed into the denatured ssRNA state.

In order to obtain atomistic insights into the unwinding mechanism, it is critical to investigate the structural dynamics of both the DDX protein and the bound RNA molecule along the unwinding reaction trajectory, as well as to quantitatively characterize the thermodynamic details underlying the binding of DDX to each RNA structural sub-state. Herein, we use solution nuclear magnetic resonance (NMR) spectroscopy that can provide atomistic insights into such dynamic DDX-RNA complexes, focusing on the core region of human DDX3X which plays an important role in the translational regulation of gene expression[3–5]. We demonstrate that preferential binding to the ssRNA state, by forming the domain-closed structure of DDX3X, serves as the driving force for remodeling structured RNAs by redistributing its structural equilibrium. These results provide the structural and thermodynamic basis underlying the physiological function of DDX proteins, enabling the remodeling of a diverse set of higher-order RNA structures.

## Results

### Characterizations of domain dynamics of DDX3X

Here we used a minimal functional core of DDX3X (residues 132-607, hereafter referred to as DDX3X), which was previously shown to exhibit dsRNA unwinding activity[19] (Fig. 1a). The activity of the purified

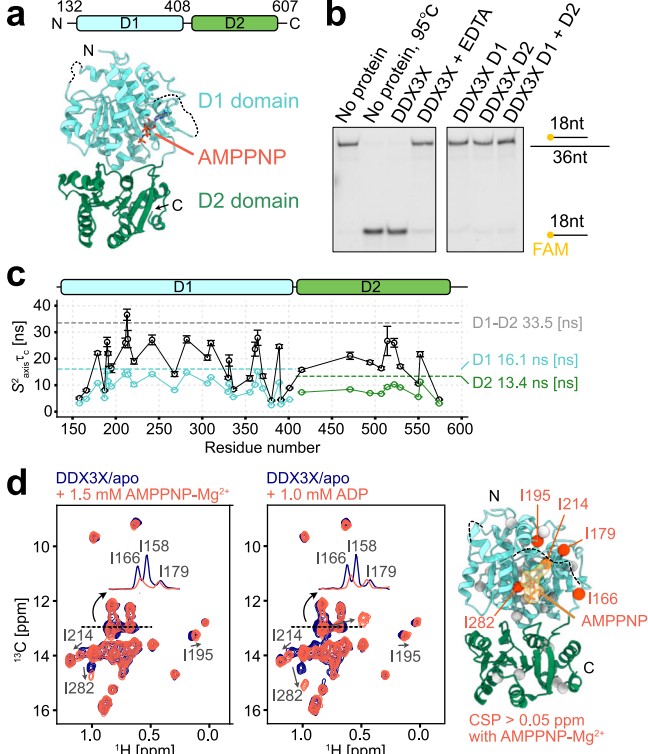

**Fig. 1 | Structure and domain dynamics of DDX3X. a** The domain architecture and the crystal structure of the core region of DDX3X in the AMPPNP-bound state are shown (PDB ID: 5E7M)[19]. The bound AMPPNP is shown as a stick model. **b** RNA unwinding assays for DDX3X and its individual domains. The upper band corresponds to the 18mer/36mer dsRNA, while the lower band corresponds to the 18mer ssRNA displaced from dsRNA by the unwinding activity of DDX3X. In the fourth lane labeled with EDTA, 20 mM EDTA was added to the reaction mixture to chelate the cofactor Mg²⁺ as a negative control. The protein concentration was 2 μM, and the reaction mixture was incubated for 30 mins at 37 °C. The assays were repeated three times with similar results. **c** Plots of $S^2_{axis}\tau_c$ for Ile and Met methyl groups of DDX3X (D1-D2, black) and its individual domains (D1 and D2, colored blue and green, respectively). Predicted values from HYDROPRO are shown as dotted lines. The measurements were performed at 35 °C. The error bars represent the uncertainties of the fitted parameters as estimated from the covariance matrix in the fitting process over 11 (for D1-D2) or 14 (for D1 and D2) data points. The center of the error bar represents the best-fit value. **d** Overlays of ¹³C-¹H HMQC spectra of [Frac-²H; Ileδ1-¹³C¹H₃; Metε-¹³C¹H₃]-labeled DDX3X with (orange-red) and without (navy) AMPPNP (left) or ADP (right). The signals with chemical shift changes are labeled. Methyl residues with chemical shift perturbations (CSPs) larger than 0.05 ppm in the presence of 1.5 mM AMPPNP are mapped onto the structure as orange spheres (PDB ID: 5E7M)[19]. I158 is located in the disordered loop whose density was not observed in the crystal structure. The measurements were performed at 35 °C and 600 MHz. Source data are provided as a Source Data file.

proteins was confirmed by an unwinding assay using a fluorescently labeled 18mer/36mer duplex as previously described[26] (Fig. 1b). The activity was validated by the increased fraction of the dissociated labeled 18mer in the presence of DDX3X, which was further confirmed by the reaction time- and protein concentration-dependent increase in the ssRNA fraction (Supplementary Fig. S1a–d). To assess the functional roles of the D1 and D2 domains, we conducted the assay using truncated proteins consisting solely of the D1 or D2 domain. Unwinding activity was not observed for these individual domains, even under conditions where both D1 and D2 proteins were added. This shows that the activity requires the presence of the intact core assembly formed by the tandem D1–D2 domains, and these two domains coordinately function to unwind dsRNA.

Since the relative orientation of the two domains is proposed to be important in binding to RNA substrates[19,22], we evaluated domain dynamics by measuring the rotational tumbling of DDX3X in the apo state under solution conditions and compared the results with data obtained from the individual D1 and D2 proteins. Since the molecular mass of DDX3X is large (53.6 kDa as a monomer), we employed the methyl transverse relaxation optimized spectroscopy (methyl-TROSY) technique and observed signals from Ile and Met side-chain methyl groups[27,28]. The $^{13}$C-$^{1}$H heteronuclear multiple quantum coherence (HMQC) spectrum of DDX3X was virtually identical to the overlay of spectra from the individual D1 and D2 domains (Supplementary Fig. S2a, b), and the assignments of DDX3X could be readily transferred from those of the individual D1 and D2 domains that were established using the standard sequential resonance assignment technique[29–32]. To evaluate the domain dynamics, we measured the $S^2_{axis}\tau_c$ of the Ile and Met methyl probes, where $S^2_{axis}$ is an order parameter that is related to the amplitude of motion of the methyl group 3-fold symmetry axis and $\tau_c$ is a rotational correlation time[33] (Fig. 1c). The $S^2_{axis}\tau_c$ values of DDX3X were distributed between 4 to 37 ns, markedly larger than those for the D1 (~2–16 ns) and D2 (~3–12 ns) domains. The $S^2_{axis}\tau_c$ value of DDX3X was on average 1.9-fold higher than those measured from the individual domains, and even up to 2.9-fold higher for the I514 methyl probe. This increase in the $S^2_{axis}\tau_c$ distribution was significantly larger than what would be expected in a scenario where two domains are connected by a short linker (~10 residues in DDX3X) without specific inter-domain interactions; in the case of tandem GB1 domains, for example, an approximately 1.4- to 1.5-fold increase in overall $\tau_c$ was observed with a linker length of 6 to 12 residues[34]. The overall $\tau_c$ value was estimated to be ~37 ns based on the highest $S^2_{axis}\tau_c$ value (for I213) assuming isotropic rotational tumbling. This value was in reasonable agreement with the calculated value of 33.5 ns from HYDROPRO[35,36] using the crystal structure of the adenosine monophosphate (AMP)-bound form of DDX3X (PDB ID: 5E7J)[19]. The $S^2_{axis}\tau_c$ values for the rest of the methyl probes were generally lower than this value, reflecting the side-chain flexibility of Ile and Met side chains whose $S^2_{axis}$ value is typically distributed around 0.1 to 0.8 and ~0 to 0.7, respectively[37]. This overall $\tau_c$ estimation suggests that the rotational tumbling of the D1 and D2 domains is to some extent restricted in the full-length DDX3X context, likely due to the formation of transient interdomain interactions even in the apo state. Additionally, these results support the notion that DDX3X predominantly exists as a monomer under our experimental conditions.

While the overall $\tau_c$ estimation is consistent with the HYDROPRO estimation based on the crystal structure, the very small chemical shift difference between the tandem D1-D2 and the individual domains (Supplementary Fig. S2a and b) strongly suggests that the interaction between these two domains is rather weak and that there is structural flexibility at the D1-D2 domain interface. To more directly characterize this inter-domain interaction, we introduced a methyl probe, M531, at the D1-D2 domain interface by replacing R531, and then measured an HMQC-nuclear Overhauser spectroscopy (NOESY) spectrum focusing on methyl-methyl contacts between these two domains. In the

modeled structure of the R531M variant based on the apo DDX3X crystal structure (PDB ID: 5E7I)[19], the side chain of M531 on the D2 domain is close to the M355 side chain on the D1 domain (Cε−Cε distance of ~5.3 Å) (Supplementary Fig. S3a), which allows us to monitor the presence of the inter-domain interaction directly through an NOE cross-peak between the M531 and M355 methyl resonances. The Met methyl signal of M531 could be readily assigned by comparing the spectra of the wild type and the R531M variant (Supplementary Fig. S3b). In the NOESY spectrum recorded with a mixing time of 500 ms, the cross-peak between these two Met resonances was not observed, indicating the presence of structural flexibility and/or heterogeneity at the D1-D2 domain interface (Supplementary Fig. S3c). We also confirmed that a cross-peak originating from the intra-domain interaction between M531 and I507 (Cε-Cδ1 distance of ~4.7 Å) was clearly observed in the NOESY spectrum, and that the overall $S^2_{axis}\tau_c$ distribution was not significantly affected in the R531M variant (Supplementary Fig. S3d). Taken together with the interpretation of the overall $\tau_c$ value, these results indicate the presence of structural flexibility at the D1-D2 interdomain interface while retaining the major domain configuration consistent with the crystal structure. Such domain flexibility likely plays a role in the recognition of the RNA substrate, as will be described in detail below.

## Localized effects upon the binding of AMPPNP or ADP

Next, we sought to investigate the interactions between DDX3X and ATP or adenosine diphosphate (ADP) to see whether the binding of ATP or ADP is coupled to domain reorganization, which has been proposed to be important in the unwinding process[19,20]. We compared $^{13}$C-$^{1}$H HMQC spectra in the absence and presence of a slowly hydrolyzable ATP analog, 5'-adenylylimidodiphosphate (AMPPNP), or ADP (Fig. 1d and Supplementary Fig. S4a). The dissociation constants of AMPPNP and ADP were estimated to be 910 ± 150 μM and 51 ± 7.9 μM, respectively, from two-dimensional NMR line-shape analyses[38] (Supplementary Fig. S4b). Upon the addition of AMPPNP or ADP, noticeable chemical shift changes or signal intensity reductions were observed only for a subset of residues: I158, I166, I179, I195, I214, and I282 which are located close to the nucleotide-binding pocket in the D1 domain[11,19], consistent with the previous study[11]. No significant chemical shift perturbations were observed in the methyl probes from the D2 domain. We also measured the apparent transverse relaxation rates of methyl $^{13}$C[39], which serve as an indicator of motion on the nanosecond to picosecond timescale and, therefore, reflect domain flexibility. We confirmed that the relaxation rates were not significantly altered by the binding to AMPPNP or ADP (Supplementary Fig. S4c). Collectively, these results demonstrate that the binding of AMPPNP or ADP induces local changes in the structure of DDX3X, while it is not strongly coupled to the domain reorganization of the D1 and D2 domains.

## DDX3X/ATP/poly-U₁₀ complex forms the closed conformation

In order to characterize DDX3X-RNA interactions, we first analyzed its interaction with ssRNA using a 10mer poly-uridine sequence (poly-U$_{10}$), which has been widely used as a model ssRNA in structural studies[15,17,21,23]. We monitored the binding of poly-U$_{10}$ by observing the Met methyl region of the $^{13}$C-$^{1}$H HMQC spectra measured in the presence of AMPPNP, ATP, or ADP (Fig. 2a). In the experiments involving ATP, we used an E348Q variant, where the glutamate from the conserved DEAD motif is replaced with glutamine, to slow down the hydrolysis of ATP. Upon the addition of poly-U$_{10}$, a new set of signals appeared in the spectra of the AMPPNP-bound and ATP-bound states, indicating that the binding of poly-U$_{10}$ induced conformational changes in DDX3X, while no such spectral changes were observed in the ADP-bound state. This shows that poly-U$_{10}$ selectively binds to AMPPNP- or ATP-bound DDX3X to form a well-defined complex with ssRNA, which is consistent with the previous studies that RNA affinity is coupled to ATP binding and that RNA is rapidly released following ATP

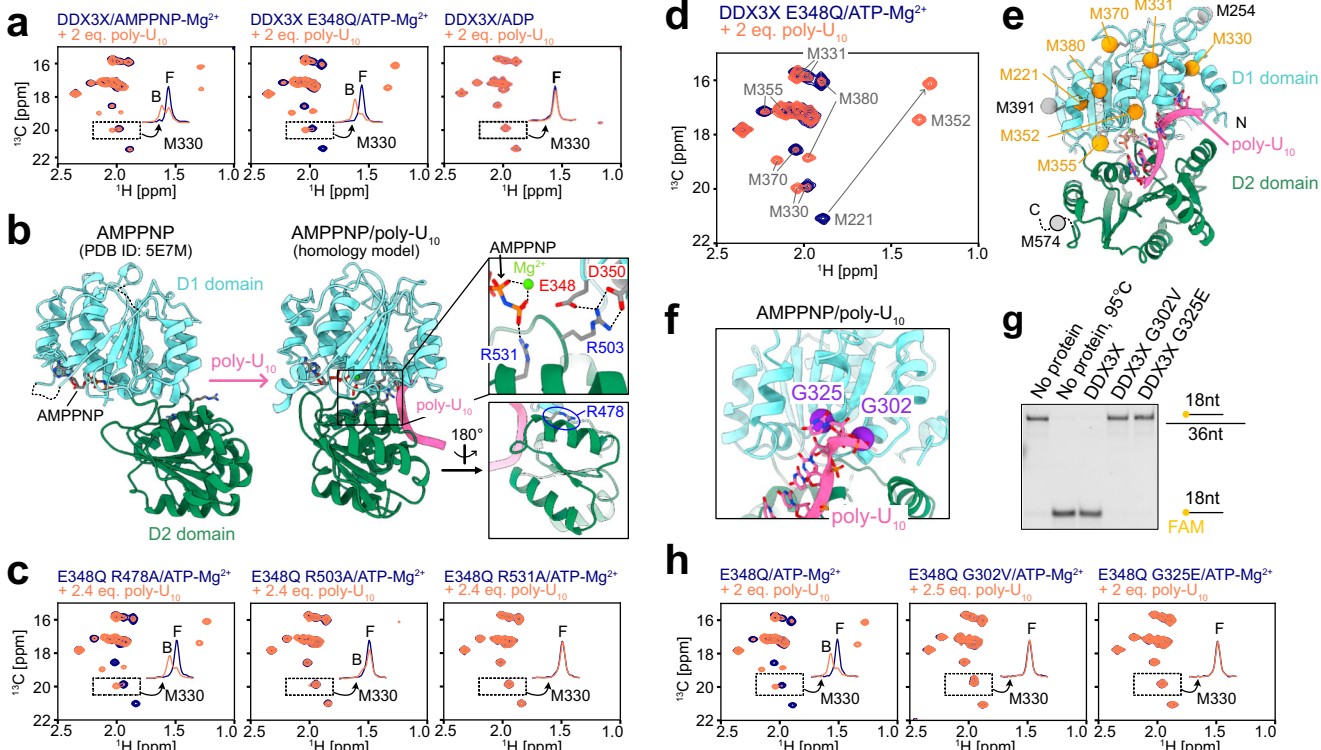

**Fig. 2 | NMR characterizations of the interaction with ssRNA. a** Overlays of $^{13}$C-$^1$H HMQC spectra of [Frac-$^2$H; Ileδ1-$^{13}$C$^1$H$_3$; Metε-$^{13}$C$^1$H$_3$]-labeled DDX3X in the absence (navy) and presence (orange-red) of poly-U$_{10}$ ssRNA measured with AMPPNP (left), ATP (center), or ADP (right). The spectrum with ATP was recorded using the E348Q variant. **b** Structural rearrangement of DDX3X upon binding to poly-U$_{10}$. The crystal structure of AMPPNP-bound DDX3X (left, PDB ID: 5E7M)[19] and the modeled structure of AMPPNP/poly-U$_{10}$-bound DDX3X (right) are shown. Close-up views of key interdomain interactions are highlighted in the insets. The mutated Arg residues are shown as stick models. **c** Overlays of $^{13}$C-$^1$H HMQC spectra of E348Q/R-to-A variants of [Frac-$^2$H; Ileδ1-$^{13}$C$^1$H$_3$; Metε-$^{13}$C$^1$H$_3$]-labeled DDX3X, measured in the absence (navy) and presence (orange-red) of poly-U$_{10}$. **d** Overlay the $^{13}$C-$^1$H HMQC spectra of [Frac-$^2$H; Ileδ1-$^{13}$C$^1$H$_3$; Metε-$^{13}$C$^1$H$_3$]-labeled DDX3X E348Q recorded with (orange-red) and without (navy) poly-U$_{10}$. **e** Mapping of methionine residues that showed significant chemical shift changes upon the binding to poly-U$_{10}$ onto the modeled structure of DDX3X/AMPPNP/poly-U$_{10}$. Methionine methyl carbons are shown as spheres. The Met methyl probes that showed significant chemical shift perturbation (CSP) are colored orange, while those with small or undefined CSP are colored gray. M574 is located in the C-terminal tail region whose structure was not modeled from the VASA/AMPPNP/poly-U$_{10}$ crystal structure (PDB ID: 2DB3). **f** Close-up view of the poly-U$_{10}$-bound structure of DDX3X showing G302 and G325 residues. Cα carbon of G302 and G325 are shown as purple spheres. **g** RNA unwinding assays for wild-type and Gly variant DDX3X using the same condition as in Fig. 1b. The assays were repeated three times with similar results. **h** Overlays of $^{13}$C-$^1$H HMQC spectra of E348Q/G302V or E348Q/G325E variants of [Frac-$^2$H; Ileδ1-$^{13}$C$^1$H$_3$; Metε-$^{13}$C$^1$H$_3$]-labeled DDX3X, measured in the absence (navy) and presence (orange-red) of poly-U$_{10}$. All NMR measurements were performed at 35 °C and 600 MHz, and the protein concentration was 40 μM (for the E348Q/G302V variant) or 50 μM (for the others). The 1D projections of the dotted region containing the free (F) and bound (B) signals for M330 are shown in each spectrum. Source data are provided as a Source Data file.

hydrolysis[40–43]. Notably, the apparent affinity for poly-U$_{10}$ was higher in the ATP-bound state of the E348Q variant compared to the AMPPNP-bound state of the wild-type protein, suggesting that the precise structural arrangement of the phosphate groups is critical for the stable formation of the ssRNA-bound state. Hereafter, we used the DDX3X E348Q variant-ATP complex for further analyses.

The most straightforward interpretation of the observed spectral changes upon binding to poly-U$_{10}$ is that DDX3X adopts the well-established closed conformation as observed in the crystal structure of the closely related *Drosophila melanogaster* VASA/AMPNNP/poly-U$_{10}$ complex[15], and in many other DDX-ssRNA complexes[21,23–25] (Fig. 2b). In this closed conformation, the side chains of conserved arginine residues (R503 and R531 in DDX3X) from the D2 domain form interdomain salt bridges with the aspartate residues in the DEAD motif (E348 and D350 in DDX3X) from the D1 domain, along with the phosphate group from the bound AMPPNP. To establish whether such interactions are formed in the poly-U$_{10}$ bound state, we analyzed poly-U$_{10}$ binding using R503A and R531A variants, and we confirmed that the poly-U$_{10}$-dependent spectral changes were abolished in these variants (Fig. 2c). Further, we verified that this change is not solely due to surface charge alterations, as poly-U$_{10}$ binding was similarly

observed in the R478A variant, where the side chain is not involved in the characteristic D1-D2 interdomain interactions. These results show that the spectral changes observed upon binding to poly-U$_{10}$ are attributed to the formation of the closed conformation as described above.

The chemical shift change upon the formation of the closed conformation was observed in virtually all Ile and Met methyl probes, consistent with the large conformational rearrangements upon binding to poly-U$_{10}$ as observed in the crystal structural analyses (Fig. 2d, e, and Supplementary Fig. S5a, b). We observed a marked chemical shift difference in the M221 and M380 methyl probes, which likely reflects the structural rearrangement of a hydrophobic cluster in the D1 domain linking the AMPPNP/ATP-binding cleft to the poly-U$_{10}$ binding site (Supplementary Fig. S5c). Since the complete set of assignments for Ile methyl probes in the poly-U$_{10}$ bound state was not available due to severe signal overlaps and broadenings, we analyzed the chemical shift perturbation of Ile methyl probes by examining the disappearance of the free state signals (Supplementary Fig. S5a, b). The marked chemical shift changes and/or signal broadenings were observed in Ile methyl probes located in both the D1 domain (I158, I166, I190, I191, I195, I214, I268, I336, I364, I389, I401) and the D2

domain (I415 and I529), while those distant from the poly-$U_{10}$ binding site did not exhibit a chemical shift change (I514 and I550). The chemical shift difference was observed not only at the poly-$U_{10}$ binding interface but also at the AMPPNP/ATP binding site, consistent with the presence of allosteric coupling between these two sites[11,44]. In addition, the poly-$U_{10}$ binding pocket within the closed structure of DDX3X, which was modeled from the VASA/AMPPNP/poly-$U_{10}$ complex[15], showed a positive electrostatic surface that complements the negative electrostatic charge of the poly-$U_{10}$ RNA (Supplementary Fig. S5d).

### The formation of the closed conformation is essential for the unwinding activity

To establish the functional role of the closed conformation, we characterized the poly-$U_{10}$ binding properties of the G302V and G325E variants, which were identified in medulloblastoma patients and are known to severely impair the function of DDX3X[7,9,11]. These glycine residues are located near the poly-$U_{10}$ binding site of the D1 domain in the closed conformation (Fig. 2f). We conducted unwinding assays on these glycine variants and found that the unwinding activity was nearly completely abolished in these variants (Fig. 2g). In the NMR spectra of these Gly variants in the presence of poly-$U_{10}$, we did not observe any chemical shift changes upon the addition of poly-$U_{10}$ (Fig. 2h). These results demonstrate that the formation of the closed conformation by binding to ssRNA is a prerequisite for dsRNA unwinding activity and that the inability to form this conformation is closely associated with the pathogenic processes resulting from these missense mutations in DDX3X.

### The affinity of DDX3X for dsRNA is weaker than that for ssRNA

Having established the ssRNA binding properties of DDX3X, we next sought to investigate its interaction with dsRNA molecules. We first focused on the interaction of DDX3X in the ATP-bound state, which exhibited a high affinity for poly-$U_{10}$ ssRNA, by using an ATPase-deficient E348Q variant of DDX3X. Considering the fact that the unwinding activity of DDX proteins is typically inversely correlated with the stability of dsRNA[44,45], we used two self-complementary dsRNA ligands with different duplex stability, GUCA-12mer (5'-GUCAG UACUGAC-3', $\Delta G^o = -17.8$ kcal/mol) and GC-14mer (5'-GGGCGGGCC CGCCC-3', $\Delta G^o = -35.0$ kcal/mol), to investigate how duplex stability affects the binding of dsRNA to DDX3X. GC-14mer shows exceptionally high stability due to its 100% GC content and has been used as a model dsRNA ligand in previous studies[11,18]. We note that the fraction of ssRNA is estimated to be below ~0.1% at micromolar concentrations. Thus, it is reasonably expected that the spectral changes upon the addition of these RNA molecules primarily reflect the binding of DDX3X to the dsRNA state.

We measured $^{13}C$-$^1H$ HMQC spectra of ATP-bound DDX3X E348Q in the presence of varying concentrations of GUCA-12mer and GC-14mer dsRNA (Fig. 3a). As a control, we first conducted a titration experiment using poly-$U_{10}$ ssRNA and obtained an apparent dissociation constant of $17 \pm 1.8$ [μM] by fitting the intensity ratio of signals from the poly-$U_{10}$ free (F) and bound (B)/closed state (Fig. 3b). When titrating with dsRNA, we observed the appearance of a distinct set of the bound state signals, indicative of binding to dsRNA; however, the affinity for dsRNA was considerably weaker than that for poly-$U_{10}$ as judged from the relative intensity of these bound state signals. When titrating with poly-$U_{10}$ ssRNA, the bound percentage exceeded ~70% upon the addition of 2 equimolar (eq.) amounts of poly-$U_{10}$, while the bound percentage remained below 40 % even with 10 eq. amounts (as a single strand) of GC-14mer. We confirmed that the intensity of the bound state signals did not change over a period of 12 h, indicating that the NMR sample had already reached an equilibrium condition (Supplementary Fig. S6). When comparing the two dsRNA ligands, the apparent affinity was much weaker for the stable GC-14mer compared to the less stable GUCA-12mer, suggesting that a less bound state is

populated with a more stable dsRNA. Intriguingly, the chemical shift of the bound state signal almost perfectly matched those from the poly-$U_{10}$-bound state, which corresponds to the closed conformation of DDX3X bound to ssRNA (Fig. 3a), unequivocally showing that DDX3X adopts the same closed conformation upon binding to dsRNA as observed in the poly-$U_{10}$ interaction. The weaker affinity for dsRNA compared to ssRNA was further confirmed from the electrophoretic mobility shift assay (EMSA) using 3'-FAM-labeled 12mer poly-uridine ssRNA (poly-$U_{12}$-FAM) and 3'-FAM-labeled GC-14mer dsRNA (GC-14mer-FAM) (Fig. 3c). Additionally, the tight binding to GC-14mer dsRNA was not observed for DDX3X in the apo (nucleotide-free) and ADP-bound states (Supplementary Fig. S7a, b), confirming that the absence of a strong interaction with dsRNA is not dependent on the nucleotide-binding status of DDX3X.

The NMR results obtained so far indicate that, in the presence of an excess amount of dsRNA, DDX3X adopts a closed conformation similar to what has been observed in its complex with ssRNA by weakly associating with the dsRNA substrate. As pointed out previously[15], this closed conformation is not compatible with dsRNA binding because one of the dsRNA strands sterically clashes with the helix in the D1 domain (residues 357 to 366, often referred to as the wedge helix) when the dsRNA structure is aligned to the bound ssRNA (Fig. 3d). Therefore, the observed closed conformation should be interpreted either as the formation of a DDX3X-ssRNA complex through binding to the pre-existing ssRNA state, or as the formation of a DDX3X-dsRNA complex accompanied by significant deformation of the dsRNA structure (i.e., DDX3X binds to a locally unfolded ssRNA region within the dsRNA ligand). To obtain further structural insights into the interaction between DDX3X and dsRNA, we then turned to the direct NMR observation of dsRNA ligands and conducted detailed thermodynamic analyses of the complex formation.

### $^{19}F$ probe for proving dsRNA-ssRNA transitions and interactions with DDX3X

To obtain direct structural evidence for the formation of the globally or locally unwound state in dsRNA bound to DDX3X, we employed $^{19}F$ NMR to observe a chemically introduced spin probe within the RNA. $^{19}F$ is highly sensitive to structural rearrangements of RNA[46-48] and is advantageous for our applications since it does not suffer from background signals from DDX3X proteins. As the $^{19}F$ spin probe, we chose to observe a site-specifically introduced ribose 2'-$^{19}F$ probe which is sensitive to whether the labeled nucleotide resides in the dsRNA or ssRNA region[46]. To test the feasibility of the ribose 2'-$^{19}F$ spin probe to detect dsRNA-ssRNA interconversion, we first used a metastable self-complementary dsRNA, UA-12mer (5'-UUUAUUAAUAAA-3'), and introduced the ribose 2'-$^{19}F$ probe at the 11th adenosine position. Due to the rather low duplex stability of UA-12mer ($\Delta G^o = -8.2$ kcal/mol), significant amounts of ssRNA are expected to be present even at tens to hundreds micromolar concentrations. In the $^{19}F$ NMR spectrum of UA-12mer at 30 °C, we observed two distinct signals at −199.7 ppm and −200.9 ppm, corresponding to the dsRNA and ssRNA states, respectively. The relative intensity ratio dramatically changed as a function of temperature, which is a hallmark of RNA hybridization reactions (Supplementary Fig. S8a). We fitted the intensity ratio of dsRNA and ssRNA signals as a function of temperature using a standard hybridization equation, yielding the enthalpy ($\Delta H^o = -73 \pm 2.5$ [kcal/mol]) and entropy ($\Delta S^o = -220 \pm 8.3$ [cal/(K·mol)]) contributions to the duplex formation (Supplementary Fig. S8b). These values were in good agreement with predictions based on nearest-neighbor parameters[49,50], $\Delta H^o = -75.3$ [kcal/mol] and $\Delta S^o = -221.4$ [cal/(K·mol)], further validating the above assignment of the $^{19}F$ signals.

Having established that the 2'-$^{19}F$ probe is sensitive to dsRNA-ssRNA structural interconversion, we next investigated the effect of DDX3X binding on the dsRNA-ssRNA equilibrium of UA-12mer. Upon titrating the ATP-bound DDX3X proteins using the ATPase-deficient

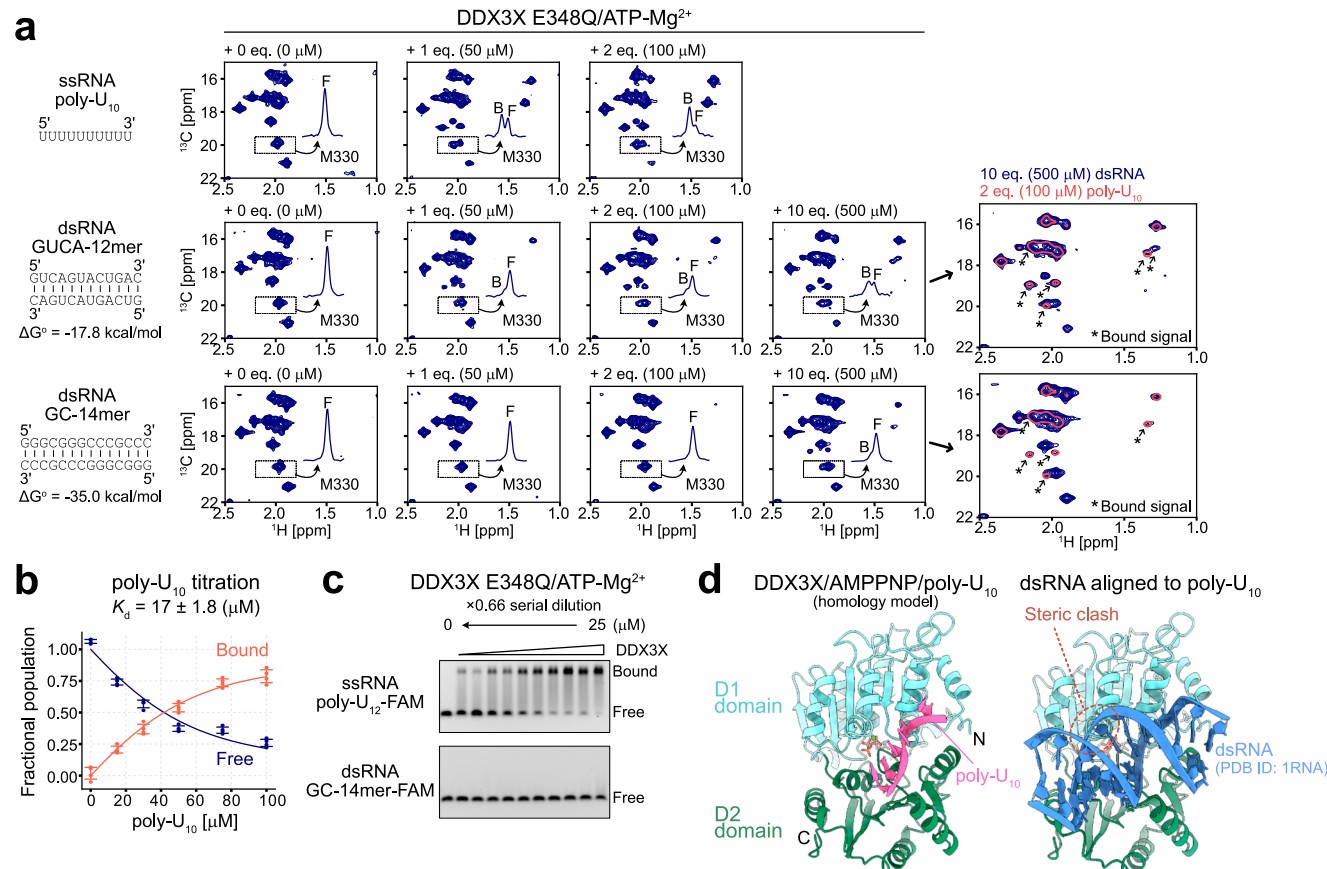

**Fig. 3 | RNA titration experiments. a** $^{13}C$-$^{1}H$ HMQC spectra of [Fracε-$^{2}H$; Ileδ1-$^{13}C^{1}H_3$; Metε-$^{13}C^{1}H_3$]-labeled DDX3X E348Q in the presence of varying concentrations of poly-U$_{10}$ ssRNA (top), GUCA-12mer dsRNA (middle), or GC-14mer dsRNA (bottom) are shown. The overlay of the spectra of DDX3X measured with 2 equimolar poly-U$_{10}$ (coral, single contour) or 10 equimolar dsRNA (navy, multiple contours) are also shown for GUCA-12mer and GC-14mer dsRNA datasets. The newly appeared bound state signals are highlighted by arrows. Duplex stability (ΔG°) at 35 °C was estimated by using nearest-neighbor parameters in 1 M NaCl[49,50]. **b** Plots of the fractional populations of the free (navy) and bound (orange-red) states as a function of poly-U$_{10}$ concentration. Experimental data points are shown as circles, and the fitted curve is shown as a line. Error bars represent the standard deviation of the data obtained from four different methyl correlations. The center of the error bar

represents the average value. **c** EMSA binding experiments for DDX3X using poly-U$_{12}$-FAM ssRNA (top) or GC-14mer-FAM dsRNA (bottom). DDX3X E348Q variant protein was titrated from 0 to 25 μM. The titration was carried out in the presence of 5 mM ATP/MgCl$_2$. The binding experiments using poly-U$_{12}$-FAM ssRNA were repeated three times with similar results, while the binding experiments using GC-14mer-FAM dsRNA were repeated twice with similar results. **d** Model structure of DDX3X bound to AMPPNP and poly-U$_{10}$ (left) or 14mer dsRNA (PDB ID: 1RNA)[82] (right). All NMR measurements were performed at 35 °C and 600 MHz, and the protein concentration was 50 μM. The 1D projections of the dotted region containing the free (F) and bound (B) signals for M330 are shown in each spectrum. Source data are provided as a Source Data file.

E348Q variant, we observed a reduction in the intensity of both dsRNA and ssRNA $^{19}F$ signals, accompanied by the appearance of a broad signal at −201.0 ppm. Notably, the chemical shift of the bound signal closely resembles that of the unbound ssRNA, indicating that UA-12mer adopts the ssRNA structure when bound to DDX3X (Fig. 4a). To further obtain the structural information of the bound form of UA-12mer, we conducted the same set of titration experiments using UA-12mer, where the $^{19}F$ spin probe was introduced by substituting the 2nd uracil base with 5-fluorouracil (5-FU) (5-FU UA-12mer)[47] (Supplementary Fig. S8c). As observed with the 2′-$^{19}F$ probe, the 5-FU chemical shift in the bound state matches that of the unbound ssRNA state. These results collectively indicate that DDX3X binds to UA-12mer to form a DDX3X-ssRNA complex.

### The binding of DDX3X leads to the unwinding of dsRNA
In order to characterize the structural changes of dsRNA upon interaction with DDX3X, we prepared the $^{19}F$-labeled GUCA-12mer and GC-14mer, where the ribose 2′-$^{19}F$ probe was incorporated at the 11th adenosine or 11th guanosine, respectively, and observed the $^{19}F$-NMR spectra in the presence of varying concentrations of the E348Q variant of DDX3X. Since these dsRNA molecules are highly stable and the

ssRNA signal could not be directly observed even at the highest accessible temperature (-60 °C), we measured the ssRNA $^{19}F$ chemical shift using reference short nucleotides that mimic the ssRNA state of GUCA-12mer and GC-14mer (Fig. 4b). Upon adding DDX3X to GUCA-12mer, a distinct broad bound signal appeared. Notably, the chemical shift of this bound state matched that of the ssRNA state, showing that GUCA-12mer unwound and formed an ssRNA-like structure upon binding to DDX3X. In the case of GC-14mer, we did not observe a distinct bound signal within the accessible concentration range of DDX3X, consistent with the result that DDX3X binds to GC-14mer less efficiently. The $^{19}F$ signal of GC-14mer showed significant broadening upon the addition of DDX3X, likely reflecting the initial encounter interaction between DDX3X and dsRNA. Significant chemical shift changes of the $^{19}F$ signal were not observed, suggesting that the broadening is due to an apparent increase in molecular weight and/or a lifetime line broadening effect[51], without the formation of a well-defined complex. This interpretation was further supported by the $^{1}H$ spectra of the imino group of GC-14mer. Specifically, the imino $^{1}H$ signals did not show site-specific chemical shift changes upon the addition of DDX3X, while an overall intensity reduction was observed accompanied by significant signal broadening (Supplementary Fig. S8d).

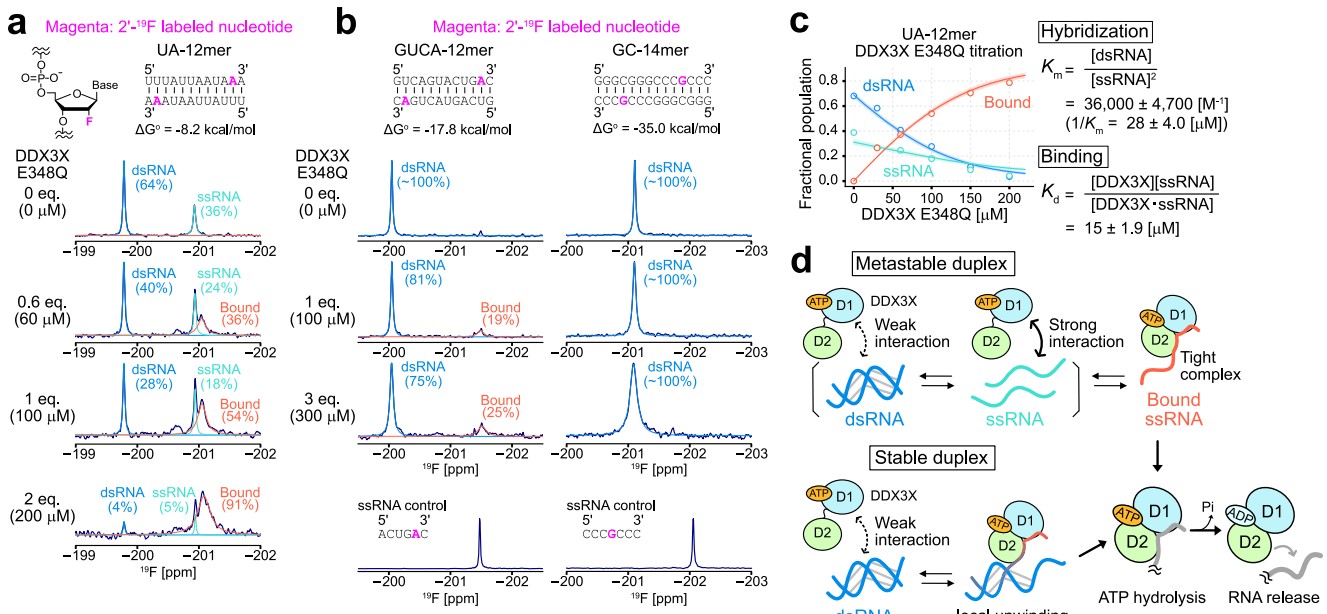

**Fig. 4 | ¹⁹F NMR analyses of RNA.** ¹⁹F 1D spectra of UA-12mer (**a**), GUCA-12mer (**b**, left), and GC-14mer (**b**, right) in the presence of varying concentrations of the E348Q variant of DDX3X in the ATP-bound form are shown. The experimentally obtained spectra are shown as navy lines, and deconvoluted lines of dsRNA (blue), ssRNA (turquoise), and bound (orange-red) signals are overlayed. The ¹⁹F NMR spectra of control ssRNA for GUCA-12mer and GC-14mer are shown below. Duplex stability ($\Delta G^o$) at 30 °C (for UA-12mer) or 35 °C (for GUCA-12mer and GC-14mer) was estimated by using nearest-neighbor parameters in 1 M NaCl[49,50]. All NMR measurements were performed at 30 °C (for UA-12mer) or 35 °C (GUCA-12mer and GC-

14mer), 600 MHz. RNA concentration was 100 μM (as a single strand). **c** Plots of the fractional populations of free dsRNA (blue), free ssRNA (turquoise), and bound ssRNA (orange-red) states as a function of DDX3X E348Q concentration. Lines represent the best-fit curves, and a 95% confidence interval of each fitted curve is contained within the thick line. The center of the error band represents the best-fit curve. The definition and best-fit value of the equilibrium constants are also displayed. **d** Cartoon representations of the interaction between each RNA ligand and DDX3X are shown. Source data are provided as a Source Data file.

## Thermodynamic basis of the stabilization of ssRNA by binding to DDX3X

The ¹⁹F NMR results indicate that the binding of DDX3X to RNA greatly stabilizes the ssRNA structure, either by binding to the pre-existing ssRNA state or by weakly associating with dsRNA initially and forming a locally unfolded complex accompanied by the deformation of the dsRNA structure. Given that the intrinsic affinity for dsRNA is much weaker than that for ssRNA, the first mechanism is appealing as a general unfolding mechanism of dsRNA. In this scenario, the NMR results can be simply interpreted as a shift in the dsRNA-ssRNA structural equilibrium towards the ssRNA state due to the tight binding of DDX3X to ssRNA, i.e. the favorable interaction between DDX3X and ssRNA leads to a predominant population of the ssRNA state within the equilibrium, which might serve as the underlying thermodynamic basis for the unwinding activity of DDX3X.

As proof of this concept, we quantitatively analyzed the UA-12mer titration profiles, considering that DDX3X binds exclusively to ssRNA for simplicity. The binding process can then be described by assuming a 3-state thermodynamic model, comprising the unbound dsRNA state, the unbound ssRNA state, and the DDX3X-bound ssRNA state. Each of these states corresponds to the three distinct signals observed in the ¹⁹F NMR spectra. These three states are related by the two equilibrium constants: $K_m$ (= [dsRNA]/[ssRNA]²) and $K_d$ (=([DDX3X][ssRNA])/ [DDX3X·ssRNA]), which describe the hybridization of UA-12mer and the association between ssRNA and DDX3X, respectively (see Methods for details). The fractional populations of each state were estimated from the signal intensities of the three states, and then the populations were fit to the above model to obtain the two equilibrium constants. The titration profiles could be well fit to the model yielding the best-fit values: $K_m = 36{,}000 \pm 4700$ [M⁻¹] (1/$K_m = 28 \pm 4.0$ [μM]) and $K_d = 15 \pm 1.9$ [μM] (Fig. 4c). Notably, the obtained $K_d$ value was in good agreement with the dissociation constant for poly-U₁₀ ( = 17 ± 1.8 [μM])

(Fig. 3b), confirming that the preferential binding to ssRNA can solely explain the titration profile of UA-12mer (Fig. 4d, top).

It is interesting to consider whether such simple mass action can also explain the ¹⁹F NMR results of GUCA-12mer. The stability of the GUCA-12mer duplex was estimated by using the nearest-neighbor parameters[49,50], yielding a $\Delta G^o$ value of −17.8 kcal/mol. This value can be recast into a $K_m$ value according to the definition of Gibbs free energy ($K_m = e^{-\Delta G^o / RT}$) to obtain a $K_m$ value of $4 \times 10^{12}$ [M⁻¹] at 35 °C (1/ $K_m = 250$ [fM]). If we consider a reaction scheme in which two DDX3X molecules globally unwind dsRNA to form two DDX3X-ssRNA complexes (2[DDX3X] + [dsRNA] → 2[DDX3X·ssRNA]), the apparent dissociation constant for this process, $K'_d$, corresponds to $K_m \times K_d^2$ (see Methods for derivation). Using the dissociation constant for poly-U₁₀ ($K_d = 17$ [μM]), the $K'_d$ value is estimated to be ~1200 [M], meaning that the affinity of DDX3X for ssRNA is not strong enough to efficiently compete with the self-hybridization reaction of GUCA-12mer. Under the ¹⁹F NMR experimental conditions, where the total concentration of GUCA-12mer is 100 μM as a single strand, the ssRNA population is expected to increase from 0.0035 % to 0.066 % by adding 300 μM ATP-bound E348Q DDX3X. However, this increase is much smaller than the observed fractional population of the bound state, which was as high as ~25% in the ¹⁹F NMR spectrum (Fig. 4b). The same discussion applies to the NMR experiments observing DDX3X methyl probes. When 500 μM (as a single strand) GUCA-12mer was added to 50 μM ATP-bound E348Q DDX3X, the bound population is calculated to be 0.046 % if we only consider the binding of DDX3X to ssRNA, which does not agree with our NMR results where the free and bound populations were comparable (Fig. 3a). These thermodynamic considerations indicate that the results obtained for stable duplexes, such as GUCA-12mer, cannot be fully explained by considering the binding to ssRNA alone. Instead, the alternative pathway involving binding to dsRNA should be included to fully account for the experimental observations.

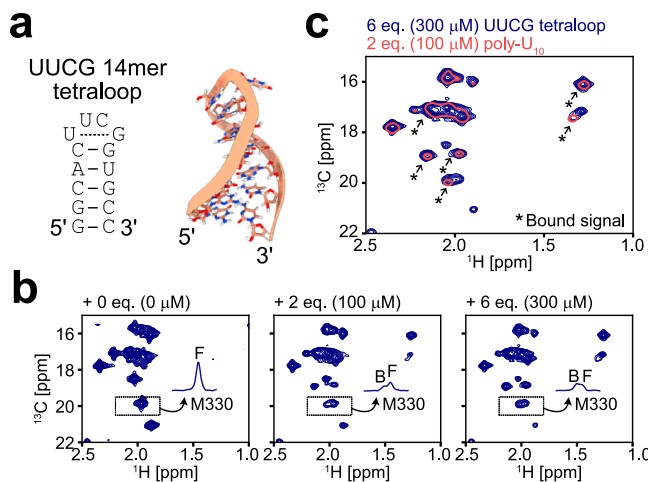

**a**

UUCG 14mer tetraloop

```
        U  C
     U        G
     C  –  G
     A  –  U
     C  –  G
     G  –  C
5' G  –  C 3'
```

**c**

6 eq. (300 µM) UUCG tetraloop
2 eq. (100 µM) poly-U$_{10}$

*Bound signal

**b**

$^{13}$C-$^1$H HMQC spectra

+ 0 eq. (0 µM)    + 2 eq. (100 µM)    + 6 eq. (300 µM)

F / M330         B F / M330          B F / M330

**Fig. 5 | UUCG tetraloop titration experiments. a** Secondary structure (left) and NMR three-dimensional structure (right) of the 14mer UUCG tetraloop (PDB ID: 2KOC)[53]. **b** $^{13}$C-$^1$H HMQC spectra of [Frac-$^2$H; Ileδ1-$^{13}$C$^1$H$_3$; Metε-$^{13}$C$^1$H$_3$]-labeled DDX3X E348Q in the presence of varying concentrations of the 14mer UUCG tetraloop. The 1D projections of the dotted region containing the free (F) and bound (B) signals for M330 are shown in each spectrum. **c** Overlay of the spectra of DDX3X measured with 2 equimolar poly-U$_{10}$ (coral, single contour) or 6 equimolar UUCG tetraloop (navy, multiple contours) is shown. The newly appeared bound state signals are highlighted by arrows. All NMR measurements were performed at 35 °C and 600 MHz, and the protein concentration was 50 µM.

We propose that the signal observed when DDX3X binds to GUCA-12mer does not reflect the global unwinding of the duplex; rather, this signal likely represents a locally unwound state of the duplex upon interaction with DDX3X, as schematically illustrated in Fig. 4d bottom. In this complex, DDX3X forms a closed conformation with ssRNA through the local unfolding of the duplex, while the ligand RNA maintains its base-paired structure without complete separation of the two strands. The formation of such a locally unwound state of dsRNA presumably represents the initial intermediate state preceding the dissociation of dsRNA.

## Formation of the DDX3X/ATP/ssRNA complex by binding to structured RNA

As a final test, we characterized the interaction between structured RNA besides dsRNA. If DDX3X recognizes the ssRNA state of structured RNA, it should be able to locally or globally unwind any structured RNA by forming the DDX3X-ssRNA complex, because DDX does not recognize base moieties in the closed conformation[15] and the structure in the ssRNA state should be common regardless of the sequence. With this in mind, we characterized the interaction with the 14mer UUCG tetraloop (5'-GGCACUUCGGUGCC-3'), which forms a stable hairpin structure[52,53] (Fig. 5a). We observed the Ile and Met methyl probes of the DDX3X E348Q variant in the ATP-bound state in the presence of varying concentrations of the 14mer tetraloop (Fig. 5b). Similarly as observed in the dsRNA titration experiments, a new set of bound signals appeared upon titrating the 14mer tetraloop, and the chemical shift of the bound state closely matched that of the ssRNA-bound closed state (Fig. 5c). These results demonstrate that the formation of the DDX3X-ssRNA complex, accompanying the global or local unwinding of structured elements, is a common characteristic in DDX3X-RNA interactions, which underlies DDX3X's unwinding activity towards a diverse set of RNA ligands.

## Discussion

The DDX family is an important class of helicase proteins that remodel RNA structures to regulate RNA metabolism. Although many different structures of DDX in complex with RNA have been solved, it has

remained unclear how the molecule recognizes and remodels a diverse set of RNA structures, as well as the underlying thermodynamic driving force for the unwinding process. In this study, we have used solution NMR to characterize the interactions between DDX3X and various RNA elements, focusing on both DDX3X and RNA conformational states, to address these questions. We demonstrated that the intrinsic affinity of DDX3X for dsRNA is markedly lower than that for ssRNA, and that the binding to ssRNA, accompanied by the formation of the closed conformation, underlies the unwinding process of dsRNA ligands. Notably, we observed a similar structural transition into the ssRNA-bound form when interacting with the 14mer UUCG tetraloop, which forms the hairpin structure, suggesting that this mechanism is at play not only in the unwinding of dsRNA but also in the remodeling of a diverse set of structured RNAs. Taken together, these results indicate that the preferential binding to ssRNA is the underlying driving force for the unwinding of structured RNAs. Since this mechanism involves the recognition of the common unstructured state of RNAs, it is compatible with the unwinding of a variety of structured RNA substrates. Moreover, we showed that impaired binding to ssRNA results in a loss of unwinding activity, ultimately leading to the onset of diseases caused by DDX3X dysfunctions.

Several studies have proposed the functional cycle for dsRNA unwinding, involving the apo DDX, the pre-unwound DDX-dsRNA complex, and the post-unwound DDX-ssRNA complex, where the domain reorganization induced by the binding and hydrolysis of ATP facilitates the unwinding of dsRNA[18,20,21]. Our NMR results demonstrated that, although there is some structural flexibility and/or heterogeneity at the D1-D2 domain interface, the domain motion is rather restricted, with the major conformation consistent with the crystal structure with the closed D1-D2 interface. Importantly, we found that the interdomain dynamics are not strongly coupled to the AMPPNP/ADP status, indicating that domain reorganization is not tightly linked to the hydrolysis of ATP during the RNA unwinding reaction. Further, we found that the inherent binding affinity for dsRNA is significantly weaker than that for ssRNA, thus the pre-unwound DDX-dsRNA complex is not largely populated during the unwinding reaction cycle. Rather, our results are more in line with the model that the unwinding of dsRNA is the consequence of the preferential binding to ssRNA either in the fully or locally unwound context; i.e. mass action drives the unwinding of dsRNA by forming the energetically stable DDX-ssRNA complex[25,54]. As noted above, such a mechanism can explain the unwinding activity regardless of the ternary structure of the RNA to be unwound, since the model assumes that DDX recognizes a common unstructured ssRNA element. It is interesting to note that the melting of structured nucleic acids by preferential binding to the denatured, single-stranded state is a well-known mechanism that was proposed many decades ago in the context of DNA destabilization by ribonuclease[55,56] or melting of DNA by bacteriophage T4 gene 32-protein[57–60], however, to our knowledge such an idea has not been commonly applied to the melting of RNA structures. Our results indicate that this classical melting mechanism is similarly at play in the remodeling of RNA structures.

In this model, the major role of ATP is to contribute to the formation of the stable DDX-ssRNA complex. In the complex, DDX adopts the well-defined closed conformation where the phosphate group of the bound ATP directly interacts with conserved arginine residues on the D2 domain along with the set of interdomain interactions between the D1 and D2 domains (Fig. 2b). Thus, these favorable interactions serve as the thermodynamic basis for the formation of the DDX-ssRNA complex. We also note that the structural heterogeneity/flexibility at the D1-D2 domain interface would facilitate the rearrangement of the D1 and D2 domains to form this closed structure. Although we mainly used the ATPase-deficient E348Q variant of DDX3X to stabilize the DDX-ssRNA complex, it is important to note that the bound ATP is rapidly hydrolyzed in the wild-type DDX3X and the bound RNA is

subsequently released in the ADP-bound state (Fig. 4d). This is also in line with the previous findings that the hydrolysis of ATP contributes to the turnover of the reaction, while it is not necessary for the unwinding activity itself[40–43].

From our NMR results in conjunction with thermodynamic considerations of binding affinities, we propose two major pathways for the unwinding of dsRNA. The first pathway involves the binding of DDX3X to pre-existing ssRNA, while the second pathway involves the binding of DDX3X to dsRNA to form a locally unfolded DDX3X-dsRNA complex (Fig. 4d). The relative contribution of each pathway depends on the balance between the affinity of DDX3X for ssRNA and the stability of the duplex, and we expect that both pathways can contribute to the unwinding process of physiological RNA substrates. The first pathway (Metastable duplex, Fig. 4d top) predominates when the stability of dsRNA is moderate and the binding affinity of DDX3X for ssRNA ($K_d$) is comparable to the affinity for the hybridization process ($K_m$). We demonstrated that the melting of UA-12mer could be nicely explained by using this model, where the $K_d$ value (=15 [μM]) was smaller than the $1/K_m$ value (=28 [μM]) (Fig. 4c) and the concentration of DDX3X was high enough to shift the equilibrium toward the ssRNA-bound state. For the melting of GUCA-12mer and GC-14mer, on the other hand, the stability of the duplex is much higher than the affinity of DDX3X for ssRNA. Therefore, the second pathway predominates the unwinding process (Stable duplex, Fig. 4d bottom). In this pathway, the binding of DDX3X can only partially melt the stable duplex, forming the locally unwound DDX3X-dsRNA complex without completely separating the two strands. This locally unwound state of dsRNA would represent the initial structural intermediate preceding the global unwinding or strand displacement of dsRNA. Intriguingly, the presence of such a complex where DDX3X binds to a locally unwound duplex is supported by the recent crystal structure of DDX3X in complex with an RNA-DNA hybrid (PDB ID: 7LIU, Enemark and Yu, to be published), which was deposited while conducting our research (Supplementary Fig. S9). In this structure, the DDX3X protein bound to ADP-BeF$_3$ (an ATP analog) interacts with the single-stranded region at the 5′-terminus of one of the strands of the RNA-DNA hybrid duplex, which retains a partially base-paired structure at the 3′-terminal side. Notably, the conformation of DDX3X in the complex is almost perfectly consistent with the closed conformation of the DDX3X/AMPPNP/poly-U$_{10}$ structural model (RMSD of 0.75Å), which is in line with our model that the formation of the closed conformation of DDX3X underlies the global/local unwinding process of duplexes or any structured RNA elements (Supplementary Fig. S9). We also note that both scenarios are consistent with the observations that the unwinding activity is inversely correlated with the stability of duplexes[44,45], because it is reasonably expected that more free ssRNA is available or the locally unfolded conformation is more easily formed in less stable duplexes.

Apparently, it looks controversial that DDX can completely displace more stable dsRNA molecules with $\Delta G^o$ values ranging from −20 to −40 kcal/mol as reported in the literature[44,45]. We note that a number of factors need to be considered to directly relate our results to these observations. First, the binding affinity of the E348Q variant used in the study most likely represents the lower limit. The inherent affinity of the wild-type DDX3X can be much higher, as the side chain of E348 participates in the interdomain interactions that stabilize the ssRNA-bound closed structure (Fig. 2b). Also, most of the in vitro unwinding assays, including ours, typically use a dsRNA substrate harboring a 5′- and/or 3′-ssRNA extension region to stimulate the unwinding activity[19,20,26,44]. The presence of these extensions would greatly facilitate the unwinding reaction both by increasing the local concentration of DDX through binding to the ssRNA extension and by allowing more than two molecules of DDX to interact with a single dsRNA ligand through the interaction at the dsRNA-ssRNA junction region, thereby contributing more to the stabilization of the DDX-ssRNA complex[61].

Consistently, the positive cooperativity involving multiple DDX molecules was observed in previous studies[20,44,62]. Wurm has recently solved the crystal structure of *E. coli* DDX DbpA bound to the dsRNA-ssRNA junction region, and shown that the binding to the junction indeed promotes the unwinding reaction[25]. Another important point to be considered is that DDX proteins can be locally concentrated around the dsRNA substrate via an accessory domain that recognizes dsRNA, which would facilitate the unwinding process as well. In the case of DDX3X, the N-terminal tail region outside the folded core likely plays this role by binding to structured RNA elements[63].

The mechanism by which DDX functions as an ssRNA binder is consistent with the diverse functional roles of the DDX family. For example, DDX is involved in RNA transport processes by forming a protein-RNA complex, where DDX serves as an immobile RNA clamp within the complex by binding to ssRNA in an ATP-dependent manner[1]. Another prominent example is the chaperone activity of DDX in assisting RNA folding[17]. The binding of DDX can reshape the folding landscape by stabilizing an unfolded ssRNA state or a locally unfolded intermediate, and help the RNA molecule to find an energetically favorable, active conformation; much like the way in which protein chaperones refold or disaggregate protein substrates by binding to unfolded proteins and/or stabilizing partially-folded intermediates[64]. Our results highlight the pivotal role of the dynamic interaction between DDX and RNA molecules and the affinity switch in response to the structural state of RNA, contributing to our understanding of the central roles of DDX in RNA metabolism.

## Methods

### Protein expression and purification

The human DDX3X core (residues 132-607, Uniprot: O00571) gene was synthesized by Eurofins Genomics and cloned into a modified pET28a(+) vector (Novagen 69864) containing the T7pCONS promoter and the translation initiation region, TIR-2[65], along with an N-terminal His$_6$-SUMO tag followed by two glycine linker residues. All DDX3X mutations, including domain truncations, were introduced by inverse polymerase chain reaction (PCR) and verified by sequencing. The DNA sequence of the DDX3X constructs and the primers used for inverse PCR are provided in Supplementary Data 1.

For producing unlabeled DDX3X proteins, transformed *E.coli* BL21(DE3) cells (ThermoFisher Scientific C600003) were grown in LB medium (MP-Biomedicals 3002-131) at 37 °C. Cells were induced by adding 1 mM isopropyl β-D-1-thiogalactopyranoside (IPTG) (Nacalai Tesque 19742-07) at an OD$_{600}$ of ~1.0 and grown for ~4 h at 37 °C. Due to the markedly lower yield of the DDX3X protein in deuterated M9 medium, we employed a fractional deuteration approach to prepare side-chain methyl–$^{13}$C$^1$H$_3$ labeled samples, except for experiments where a high level of deuteration is preferred[66,67]. For producing fractionally (Frac-)/uniformly(U-) $^2$H-labeled proteins, the transformed *E.coli* BL21(DE3) cells were grown in minimal M9 H$_2$O/D$_2$O (Cambridge Isotope Laboratories DLM-4) media supplemented with 3.0 g/L [U-$^2$H] D-glucose (Cambridge Isotope Laboratories DLM-2062) and 0.5 g/L [U-$^2$H] algal amino acid mixture (Cambridge Isotope Laboratories DLM-2082). For selective Ileδ1 and Metε methyl $^{13}$C$^1$H$_3$-labeling, 120 mg/L 2-ketobutyric acid-4-$^{13}$C,3,3,-d2 (ISOTEC 589276) and 100 mg/L L-methionine-(methyl-$^{13}$C) (ISOTEC 299146) or [2,3,3,4,4,-d5; methyl-$^{13}$CH$_3$] L-methionine (Cambridge Isotope Laboratories CDLM-8885) were added 1 h before induction of protein overexpression[68,69]. Cells were induced by adding 1.0 mM IPTG at an OD$_{600}$ of ~1.0 and grown for ~4 h at 37 °C. Proteins were purified by Ni$^{2+}$-affinity chromatography using a Ni-NTA Agarose resin (QIAGEN 30230). The resin was extensively washed using a buffer containing 1 M NaCl (Nacalai Tesque 31320-05) to remove bound nucleic acids[19], and proteins were eluted in a buffer containing 20 mM Tris-HCl (pH 8.0) (Nacalai Tesque 35434-21), 300 mM NaCl, and 300 mM imidazole (Nacalai Tesque 08787-35). The N-terminal His$_6$-SUMO tag was cleaved by the addition

of Ulp1 protease. The proteins were further purified by size exclusion chromatography on a Superdex™ 200 Increase column (Cytiva 28990944) in a buffer containing 20 mM Tris-HCl (pH 8.0), 300 mM NaCl, and 1 mM dithiothreitol (DTT) (Nacalai Tesque 14112-94). Since DDX3X proteins tend to aggregate under low salt conditions (<300 mM) in the absence of nucleotides, the proteins were initially concentrated in the presence of 300 mM NaCl using an Amicon® Ultra MWCO30K (Merck UFC903024) concentrator and diluted into a low-salt buffer with or without desired nucleotides. The protein concentration was estimated based on absorbance at 280 nm using a molar extinction coefficient of 40,340 $M^{-1}$ $cm^{-1}$.

For producing unlabeled DDX3X D1 (residues 132-408) and D2 (residues 409-607) proteins, transformed *E.coli* BL21(DE3) cells were grown in LB medium at 37 °C. Cells were induced by adding 0.4 mM IPTG at an $OD_{600}$ of ~1.0 and grown for ~12 h at 25 °C. For producing [U-$^2$H, $^{13}$C, $^{15}$N; Ileδ1-$^{13}$C$^1$H$_3$; Leuδ/Valγ-$^{13}$C$^1$H$_3$/$^{12}$CD$_3$]-labeled DDX3X D1/D2 proteins, the transformed *E.coli* BL21(DE3) cells were grown in minimal M9 D$_2$O media supplemented with 3.0 g/L [U-$^2$H] D-glucose and 1.0 g/L [U-$^{15}$N] ammonium chloride (Cambridge Isotope Laboratories NLM-467) as the sole carbon and nitrogen source, respectively, and 80 mg/L [1,2,3,4-$^{13}$C4;3,4',4',4',-D4] alpha-ketoisovaleric acid (Cambridge Isotope Laboratories CDLM-8100) and 60 mg/L [$^{13}$C4; 3,3-D2] alpha-ketobutyric acid (Cambridge Isotope Laboratories CDLM-4611) was added 1 h before induction of protein overexpeession. For producing [U-$^2$H; Ileδ1–$^{13}$C$^1$H$_3$; Metε-$^{13}$C$^1$H$_3$] proteins, cells were grown in minimal M9 D$_2$O media supplemented with 3.0 g/L [U-$^2$H] D-glucose, and 60 mg/L 2-ketobutyric acid-4-$^{13}$C,3,3-d2 and 100 mg/L L-methionine-(methyl–$^{13}$C) were added 1 h before induction of protein overexpression. Cells were induced by adding 0.4 mM IPTG at an $OD_{600}$ of ~1.0 and grown for ~16 h at 25 °C. Proteins were purified using the same protocol as for the purification of the DDX3X core, described above. The protein concentration was estimated based on absorbance at 280 nm using a molar extinction coefficient of 20,400 $M^{-1}$ $cm^{-1}$ and 19,940 $M^{-1}$ $cm^{-1}$ for the D1 and D2 domains, respectively.

## RNA preparation
All RNA oligos except those with 5-FU modification were synthesized by Hokkaido System Science Co., Ltd. RNA oligos with 5-FU modification were synthesized by Dharmacon Inc. as a protected form. Deprotection was performed following the manufacturer's instructions. The 14mer UUCG tetraloop was refolded as previously described[52]. Briefly, the RNA was first denatured for 10 mins at a concentration of 250 μM and 95°C, followed by 20-fold dilution with ice-cold water. The duplex stability (ΔG°) was estimated by the nearest-neighbor approach using the parameters in 1.0 M NaCl[49,50] (https://www.konan-fiber.jp/hp/sugimoto/contents/NN/NearestNeighborCalculator.htm). The presence of the ribose 2′-F modification was not considered in the calculations of ΔG°.

## RNA unwinding assay
The RNA unwinding activity of DDX3X was monitored by following the displacement of the fluorescently labeled 18mer strand from the complementary 36mer[26]. The dsRNA substrate was prepared by annealing a 5′-FAM (6-Carboxyfluorescein) labeled 18mer RNA (5′-CCCAAGAACCCAAGGAAC-3′) and an unlabeled 36mer RNA (5′-ACCAGCUUUGUUCCUUGGGGUUCUUGGGGAGCAGCAGG-3′, wherein the underlined region is complementary to the labeled 18mer). To prevent reannealing of the labeled strand, an excess amount of unlabeled 18mer trap ssRNA was included in the reaction mixture. The reaction mixture comprised 0 to 2 μM DDX3X protein, 10 nM dsRNA, 200 nM trap ssRNA, 20 mM HEPES-NaOH (pH 7.4) (Nacalai Tesque 17546-05), 50 mM NaCl, 5 mM ATP (Nacalai Tesque 08886-64), 5 mM MgCl$_2$ (Nacalai Tesque 20908-65), 5 mM DTT, and 5% glycerol (Nacalai Tesque 17017-93). Reactions were started by adding an ATP-Mg stock solution, followed by incubation at 37 °C. At each time point, aliquots

were collected and mixed with a sodium dodecyl sulfate solution (final 0.1 %) (Nacalai Tesque 02873-75) to quench the reaction. A control sample was prepared by heating the aliquot at 95 °C for 3 min to completely displace dsRNA. The samples were separated by native PAGE on a 20% polyacrylamide gel at 150 V for 90 min in Tris-Glycine buffer (25 mM Tris, 192 mM glycine) (Nacalai Tesque 17141-95). The gel was imaged on LuminoGraph I equipped with WSE-5600 CyanoView (excitation wavelength at 505 ± 25 nm) (ATTO) and analyzed by using CS Analyzer4 (version 2.4.5) (ATTO).

## Electrophoretic mobility shift assay
The binding affinity was assessed by monitoring the band shift of fluorescently labeled RNA upon complex formation with DDX3X[70]. 3′-FAM labeled GC-14mer dsRNA (5′-GGGCGGGCCCGCCC-3′) or 3′-FAM labeled 12mer poly-uridine ssRNA was used. The reaction mixture comprised 0 to 25 μM DDX3X protein, 100 nM (as a single strand) fluorescently labeled RNA, 20 mM HEPES-NaOH (pH 7.4), 100 mM NaCl, 5 mM DTT, and 10 % glycerol. 5 mM ATP /MgCl$_2$ or 5 mM ADP (Nacalai Tesque 01652-24)/EDTA (Nacalai Tesque 15111-45) was added to measure the affinity for ATP- or ADP-bound DDX3X, respectively. The samples were separated by 2% agarose gel at 100 V for 25 min in TBE buffer (89 mM Tris-borate, 2 mM EDTA) (Nacalai Tesque 35440-44). The gel was imaged on LuminoGraph I equipped with WSE-5600 CyanoView (excitation wavelength at 505 ± 25 nm) (ATTO) and analyzed by using CS Analyzer4 (version 2.4.5) (ATTO).

## NMR experiments
All $^1$H-detected NMR measurements were performed at 35 °C unless indicated otherwise, using Bruker AVANCE-III HD or Bruker AVANCE NEO spectrometers with a cryogenically cooled $z$ pulsed-field gradient triple-resonance TCI probe or a cryogenically cooled $z$ pulse-filed gradient quadruple-resonance QCI-P probe at 600 MHz, 900 MHz, or 1 GHz. $^{19}$F-detected NMR measurements were performed at 30 °C (for UA-12mer) or 35 °C (for GUCA-12mer and GC-14mer) unless indicated otherwise, using Bruker AVANCE-III HD equipped with a cryogenically cooled $z$ pulse-filed gradient quadruple-resonance QCI-F probe at 600 MHz. All NMR data were acquired using Bruker TopSpin 3.5.7 or 4.2.0. All spectra were processed using the NMRPipe suite of programs[71] (version 11.1), analyzed by NMRFAM-SPARKY[72] (version 1.470), and visualized using the Python package nmrglue[73] (version 0.90). Peak intensities were extracted either by using the Peakipy software package (https://github.com/j-brady/peakipy) (version 0.1.30) or by ellipsoidal sum integration. The two-dimensional lineshape analyses for estimating the binding affinity of AMPPNP or ADP were performed by using TITAN software (version 1.6)[38].

Backbone resonance assignments of [U-$^2$H, $^{13}$C, $^{15}$N; Ileδ1-$^{13}$C$^1$H$_3$; Leuδ/Valγ–$^{13}$C$^1$H$_3$/$^{12}$CD$_3$]-labeled D1 and D2 domains were obtained by TROSY-based 3D triple-resonance experiments[29,30], including HNCO, HN(CA)CO, HNCA, HN(CO)CA, HNCACB, and HN(CO)CACB. The assignments of D1 were reported previously[11]. Side-chain Ile methyl assignments were obtained by recording out-and-back 3D HMCM(CG)CB and HMCM(CGCB)CA[31,32] datasets. Met methyl assignments of DDX3X in the RNA-free state were obtained by HMQC ($^{13}$C-$t_1$)-NOESY-HMQC ($^{13}$C-$t_2$, $^1$H-$t_3$) (300 ms mixing time) based on the crystal structure of DDX3X[19]. Methyl assignments were further confirmed by a pair of TROSY-based 3D NOE experiments (250-300 ms mixing time, $^1$H($t_1$)-NOE-$^{15}$N($t_2$)-$^1$H($t_3$) and $^{13}$C($t_1$)-NOE-$^{15}$N($t_2$)-$^1$H($t_3$)). Met methyl assignments of DDX3X in the poly-U$_{10}$-bound state were obtained by the mutagenesis approach. $^{13}$C-$^1$H HMQC spectra were recorded of the following assignment variants in the E348Q background; M187I, M221A, M221I, M254I, M330I, M331I, M352I, M355I, M370I, M379I, M380I, M391I, and M574I.

$^{13}$C-$^1$H HMQC spectra of [Frac-/U-$^2$H; Ileδ1-$^{13}$C$^1$H$_3$; Metε-$^{13}$C$^1$H$_3$]-labeled DDX3X were recorded using a sequence that exploits the

methyl-TROSY effect[27,74]. DDX3X sample concentration ranged from 20 to 50 µM in an NMR buffer consisting of 20 mM HEPES-NaOH (pD 7.4), 100 mM NaCl, 5 mM DTT in $D_2O$. To observe ATP-, AMPPNP-, or ADP-bound DDX3X, 5 mM ATP/$MgCl_2$, 5 mM AMPPNP (Nacalai Tesque 01070-44) /$MgCl_2$, or 5 mM ADP/EDTA was respectively added to the solution. For samples containing RNAs, 330 unit/mL RNAsin® plus RNAase inhibitor (Promega N2611) was directly added to the solution.

$S^2_{axis}\tau_c$ values of side-chain methyl groups were obtained by fitting the ratios of signal intensities measured from spectra quantifying sums ($I_{SQ}$) and differences ($I_{3Q}$) of single quantum methyl $^1H$ magnetization components as a function of a relaxation delay ($T$)[33]:

$$\left|\frac{I_{3Q}}{I_{SQ}}\right| = \frac{0.75\eta\tanh\left(T\sqrt{\eta^2+\delta^2}\right)}{\sqrt{\eta^2+\delta^2} - \delta\tanh\left(T\sqrt{\eta^2+\delta^2}\right)} \quad (1)$$

In Eq. (1). the intra-methyl $^1H$–$^1H$ dipolar cross-correlated relaxation rate, $\eta$, is given by

$$\eta \approx \frac{9}{10}\left(\frac{\mu_0}{4\pi}\right)^2 \left[P_2(\cos\theta_{axis,HH})\right]^2 \frac{S^2_{axis}\gamma_H^4\hbar^2\tau_c}{r_{HH}^6} \quad (2)$$

where $\gamma_H$ is the gyromagnetic ratio of a $^1H$ spin, $r_{HH}$ is the distance between methyl protons (1.813 Å), $P_2(x) = \frac{1}{2}(3\cos^2 x - 1)$, $\hbar$ is Planck's constant divided by $2\pi$, and $\theta_{axis,HH}$ (= 90º) is the angle between a vector connecting pairs of methyl protons and the methyl 3-fold symmetry axis. The measurements were performed at 900 MHz (for DDX3X) or 1 GHz (for D1 and D2) using a 120 µM [U-$^2H$; Ileδ1-$^{13}C^1H_3$; Metε-$^{13}C^1H_3$]-labeled sample in an NMR buffer consisting of 20 mM HEPES-NaOH (pD 7.4), 300 mM NaCl, 10 mM $MgCl_2$, 5 mM DTT in $D_2O$. The measurements of $S^2_{axis}\tau_c$ values for the R531M variant were performed at 1 GHz using a 120 µM [U-$^2H$; Ileδ1-$^{13}C^1H_3$; Met αβγ-$^2H$, ε-$^{13}C^1H_3$]-labeled sample in an NMR buffer consisting of 20 mM HEPES-NaOH (pD 7.4), 300 mM NaCl, 5 mM DTT in $D_2O$. The rotational correlation time of DDX3X and its individual domains was predicted from the crystal structure using WinHydroPro[35,36] (version 1.00).

$^{13}C$ single-quantum transverse relaxation rates were measured using a Carr-Purcell-Meiboom-Gill (CPMG) pulse scheme as described previously[39]. A CPMG field of 2 kHz was used with a constant-time relaxation delay of 15 ms. The measurements were performed at 900 MHz using a 150 µM [Frac-$^2H$; Ileδ1-$^{13}C^1H_3$; Metε-$^{13}C^1H_3$]-labeled DDX3X sample in an NMR buffer consisting of 20 mM HEPES-NaOH (pD 7.4), 300 mM NaCl, 5 mM DTT in $D_2O$. 5 mM AMPPNPP/$MgCl_2$ or 5 mM ADP/EDTA was added to observe AMPPNP- or ADP-bound DDX3X, respectively.

HMQC-($^{13}C$-$t_1$)-NOESY ($^1H$-$t_2$) (500 ms mixing time) spectrum was recorded at 1 GHz using a 220 µM [U-$^2H$; Ileδ1-$^{13}C^1H_3$; Met αβγ-$^2H$, ε-$^{13}C^1H_3$]-labeled R531M DDX3X sample in an NMR buffer consisting of 20 mM HEPES-NaOH (pD 7.4), 300 mM NaCl, 5 mM DTT in $D_2O$.

$^{19}F$ 1D NMR spectra were measured using a triple-pulse excitation scheme to suppress the residual $^{19}F$ background in the probe as well as radio-frequency acoustic ringing[75]. $^1H$ GARP-4 decoupling element[76] with a field of 1 kHz was applied during the acquisition period. A 90.6 ms acquisition time and 2 sec interscan delay was used. All time domain data were processed with an exponential decay function using a line broadening factor of 10 Hz prior to Fourier transformation. The deconvolution was performed by fitting the experimental curve by using an overlay of Lorentzian lineshape functions, wherein the amplitude, linewidth and peak-top position were varied as a fit parameter. The measurements were performed at 600 MHz using a 100 µM (as a single strand) $^{19}F$-labeled RNA sample in an NMR buffer consisting of 20 mM HEPES-NaOH (pH 7.4), 100 mM NaCl, 5 mM ATP, 5 mM $MgCl_2$, 5 mM DTT, 330 unit/mL RNAsin® plus, and 5 % $D_2O$.

## Hybridization of RNA

The hybridization reaction of self-complementary ssRNA to form dsRNA can be described using the following dimerization scheme:

$$2[ssRNA] \rightleftharpoons [dsRNA] \quad (3)$$

where [ssRNA] and [dsRNA] denote the molar concentrations of ssRNA and dsRNA, respectively. The equilibrium constant of this process, $K_m$, is given by

$$K_m = \frac{[dsRNA]}{[ssRNA]^2} \quad (4)$$

, and the total RNA concentration, $L_T$, is defined as follows:

$$L_T = [ssRNA] + 2[dsRNA] \quad (5)$$

Then, each concentration term can be obtained by solving the above equations:

$$[ssRNA] = \frac{-1 + \sqrt{1 + 8L_T K_m}}{4K_m}$$
$$[dsRNA] = K_m[ssRNA]^2 \quad (6)$$

The fractional populations of ssRNA and dsRNA, $F_{ssRNA}$ and $F_{dsRNA}$, are given by

$$F_{ssRNA} = [ssRNA]/L_T$$
$$F_{dsRNA} = 2[dsRNA]/L_T \quad (7)$$

These values are directly related to the populations estimated from NMR signal intensities.

In the thermal melting experiment, the temperature dependence of $K_m$ was evaluated from the $F_{ssRNA}$ and $F_{dsRNA}$ values obtained from the $^{19}F$ NMR spectrum at each temperature. By definition, $K_m$ at absolute temperature $T$, $K_m(T)$, is related to a standard state Gibbs free energy, $\Delta G^o$, by

$$K_m(T) = e^{-\Delta G^o/RT} \quad (8)$$

where $R$ is the gas constant (1.987 cal/(mol·K)). $\Delta G^o$ is a function of temperature and can be described using enthalpic and entropic contributions:

$$\Delta G^o = \Delta H^o - T\Delta S^o \quad (9)$$

where $\Delta H^o$ is the enthalpic change and $\Delta S^o$ is the entropic change of the duplex formation. The fractional populations of dsRNA and ssRNA states were evaluated at 22.5, 25, 27.5, 30, 32.5, 35, 37.5, 40, 42.5, and 45 °C, where signals from both states could be reliably quantified. These populations were fit to the above equations to obtain $\Delta H^o$ and $\Delta S^o$ as fit parameters.

## Thermodynamics of RNA binding to DDX3X

The binding of poly-U$_{10}$ ssRNA to DDX3X was analyzed by assuming a simple one-site binding model as follows:

$$[DDX3X] + [ssRNA] \rightleftharpoons [DDX3X \cdot ssRNA] \quad (10)$$

where [DDX3X], [ssRNA], and [DDX3X·ssRNA] denote the molar concentrations of DDX3X in the free state, unbound ssRNA, and DDX3X bound to ssRNA, respectively. The dissociation constant for the

binding of ssRNA to DDX3X is given by

$$K_d = \frac{[DDX3X][ssRNA]}{[DDX3X \cdot ssRNA]} \qquad (11)$$

and the total protein concentration, $C_T$, and total ligand ssRNA concentration, $L_T$, are given by

$$C_T = [DDX3X] + [DDX3X \cdot ssRNA]$$
$$L_T = [ssRNA] + [DDX3X \cdot ssRNA] \qquad (12)$$

Then, each concentration term can be readily calculated by using $K_d$, $C_T$, and $L_T$ as follows:

$$[ssRNA] = \frac{-C_T + L_T - K_d + \sqrt{(C_T - L_T + K_d)^2 + 4K_d L_T}}{2}$$
$$[DDX3X] = C_T - L_T + [ssRNA] \qquad (13)$$
$$[DDX3X \cdot ssRNA] = L_T - [ssRNA]$$

The NMR signal intensities in the free and bound states are directly proportional to [DDX3X] and [DDX3X·ssRNA], respectively, with an appropriate scaling factor that converts molar concentration into NMR signal intensity. The $K_d$ value was determined by globally fitting the signal intensities from M221, M355, M370, and M380 for the free state and those from M221, M352, M370, and M380 for the bound state that were measured in the absence or presence of varying concentrations of poly-$U_{10}$. During the fitting procedure, the maximum signal intensity in each state was treated as a fit parameter. Distributions of fitted parameters were obtained by running 1,000 Monte-Carlo simulations, and the errors were taken as the standard deviations of these parameter sets.

The binding of DDX3X to RNA in the presence of an equilibrium between ssRNA and dsRNA states can be described as follows:

$$2[ssRNA] \rightleftharpoons [dsRNA]$$
$$[DDX3X] + [ssRNA] \rightleftharpoons [DDX3X \cdot ssRNA] \qquad (14)$$

where [ssRNA], [dsRNA], [DDX3X], and [DDX3X·ssRNA] represent the molar concentrations of unbound ssRNA, unbound dsRNA, unbound DDX3X, and DDX3X bound to ssRNA, respectively. As described in the main text, we assume that DDX3X can bind to and form a stable complex exclusively with ssRNA. The binding equilibria can be described by using two equilibrium constants, $K_m$ and $K_d$, as defined in Eqs. (4) and (11). The total DDX3X and RNA concentrations, $C_T$ and $L_T$, are defined as

$$C_T = [DDX3X] + [DDX3X \cdot ssRNA]$$
$$L_T = [ssRNA] + 2[dsRNA] + [DDX3X \cdot ssRNA] \qquad (15)$$

The fractional populations of the unbound ssRNA ($F_{ssRNA}$), unbound dsRNA ($F_{dsRNA}$), and ssRNA bound to DDX3X ($F_{DDX3X \cdot ssRNA}$) are given by

$$F_{ssRNA} = [ssRNA]/L_T$$
$$F_{dsRNA} = 2[dsRNA]/L_T \qquad (16)$$
$$F_{DDX3X \cdot ssRNA} = [DDX3X \cdot ssRNA]/L_T$$

These values are directly related to the populations estimated from NMR signal intensities. The DDX3X titration profile was analyzed to obtain the above fractional populations at each DDX3X concentration, which were subsequently fit to obtain $K_m$ and $K_d$ values. The fitting was performed by a nested minimization procedure. In the first minimization, $C_T$ and $L_T$ values along with initial estimates of the dissociation constants were passed into the root-finding algorithm of Python 3.7 SciPy 1.3 library (scipy.optimize.root) to determine the

concentration terms ([ssRNA], [dsRNA], [DDX3X], and [DDX3X·ssRNA]). This was achieved by solving systems of equations relating protein concentrations to equilibrium constants and $C_T$ and $L_T$ values as defined in Eqs. (4), (11), (15). In the second minimization step, the extracted concentrations were used to calculate fractional populations for the three states (Eq. 16), which were then compared with the NMR signal intensities from each titration point. Distributions of fitted parameters were obtained by running 1,000 Monte-Carlo simulations, and the errors were taken as the standard deviations of these parameter sets.

In the main text, we also considered the scenario where two DDX3X molecules interact with a single dsRNA ligand to form two DDX3X·ssRNA complexes. The binding scheme is described as follows:

$$2[DDX3X] + [dsRNA] \rightleftharpoons ([2 \cdot DDX3X \cdot dsRNA]) \rightleftharpoons 2[DDX3X \cdot ssRNA] \qquad (17)$$

where the central [2·DDX3X·dsRNA] represents an intermediate state where two molecules of DDX3X transiently associate with dsRNA. Here, we assumed that the binding of two DDX3X molecules occurs with maximum cooperativity for simplicity. Assuming that the population of the intermediate state is negligibly small, the apparent dissociation constant for this process, $K'_d$, is related to $K_m$ and $K_d$ as follows:

$$K'_d = \frac{[DDX3X]^2 [dsRNA]}{[DDX3X \cdot ssRNA]^2} = \frac{[dsRNA]}{[ssRNA]^2} \cdot \left( \frac{[DDX3X][ssRNA]}{[DDX3X \cdot ssRNA]} \right)^2 = K_m K_d^2 \qquad (18)$$

### Structural modeling
The structure of DDX3X bound to poly-$U_{10}$ was modeled using the SWISS-MODEL web server (https://swissmodel.expasy.org/)[77]. The structure of the VASA/AMPPNP/poly-$U_{10}$ complex (PDB ID: 2DB3)[15] was used as a template. The surface charge of the DDX3X complex was calculated using the PDB2PQR (version 3.6.1) and APBS (version 3.4.1) software[78]. The structures presented in the figures were visualized using the UCSF ChimeraX software (version 1.6.1)[79].

### Reporting summary
Further information on research design is available in the Nature Portfolio Reporting Summary linked to this article.

## Data availability
The data supporting the findings of this study are available from the corresponding authors upon request. NMR assignments for the D1 and D2 domains of DDX3X have been deposited in the BMRB database under accession numbers 52143 and 52142, respectively. The structure data used in this study are available in the Protein Data Bank under accession codes 1RNA, 2DB3, 2KOC, 5E7I, 5E7J, 5E7M, and 7LIU. The protein sequence used in this study is available from the UniProt database under accession code O00571 (DDX3X). Source data are provided with this paper.

## Code availability
The Python scripts used in this study along with the relevant data are available on https://github.com/YukiToyama/RNA_binding_property_of_DDX3X (https://doi.org/10.5281/zenodo.10826902)[80].

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

## Acknowledgements

This work was supported by the Japan Agency for Medical Research and Development (AMED) under Grant Number JP21ae0121028 for I.S. The authors thank Drs. Koh Takeuchi, Shunsuke Imai, Yuji Tokunaga, and Yutaro Shiraishi for helpful discussions. This study made use of NMRbox[81]: National Center for Biomolecular NMR Data Processing and Analysis, a Biomedical Technology Research Resource (BTRR), which is supported by NIH grant P41GM111135 (NIGMS). Molecular graphics and analyses performed with UCSF ChimeraX, developed by the Resource for Biocomputing, Visualization, and Informatics at the University of California, San Francisco, with support from National Institutes of Health R01-GM129325 and the Office of Cyber Infrastructure and Computational Biology, National Institute of Allergy and Infectious Diseases.

## Author contributions

Y.T. conceived the project, purified the proteins, performed the biological assays, prepared the NMR samples, conducted the NMR experiments, and analyzed the data. Y.T. and I.S. designed the research and wrote the manuscript.

## Competing interests

The authors declare no competing interests.
