## [Peer Review File · Nature Communications]

NMR characterization of RNA binding property of the DEAD-box RNA helicase DDX3X and its implications for helicase activityREVIEWER COMMENTS

Reviewer #1 (Remarks to the Author):

The manuscript of Toyama and Shimada uses methyl and fluorine NMR spectroscopy to obtain insights into the mechanism of the unwinding process of the DDX3 DEAD box helicase. The work is of high quality and well presented. I do, however, have some major issues with the interpretation of the data.

My mayor point is that I disagree with the model that the authors bring up, where the helicase interacts with the dsRNA directly. Rather, the data indicate that the helicase binds to a (small) ssRNA fraction (that is in the dsRNA). This small ssRNA fraction then interacts with the helicase. As a consequence the dsRNA:ssRNA equilibrium shifts further towards the ssRNA side.

In case dsRNA would be recognized directly the affinity for any dsRNA would be very similar (the interaction is sequence independent). However, this is something that the authors clearly do not observe. In my opinion there is no (relevant) direct interaction between dsRNA and the helicase. The authors also mention this scenario (page 15: "The ^{19}F NMR results indicate that the binding of DDX3X to RNA greatly stabilizes the ssRNA state, either by binding to the pre-existing ssRNA state and/or by binding to dsRNA to form the ssRNA-DDX3X structure."). The latter option " by binding to dsRNA" does not take place. The dsRNA first (locally or globally) unwinds, after which the helicase interacts with the ssRNA (region).

As the data in the paper is in general of high quality my suggestion would be to reinterpret all data and to rephrase the text in the light of a model in which the helicase does not interact with dsRNA at all (which is also in agreement with the literature) and that helicase activity is only due to the interaction between a small fraction of ssRNA which then results in the slow unwinding of the dsRNA due to a shift in the dsRNA-ssRNA equilibrium.

In addition:

Figure 1b: The temperature should be mentioned in the legend. This is now only done for the 95C control.

Figure 1c: I am not convinced if the discussion regarding residual domain interactions based on the S_2 tauc values is conclusive. The tauc value that is discussed for the FL protein (37 ns) is an overestimation and based on one "outlier" resonance (I213 likely). In case that residue is discarded the estimate of tauc would drop to around 27 ns, smaller than what is expected from a complex with two domains that form a stable interface (thus proving that the domains are not associating in the apo state). In addition, there are no CSPs between the individual domains and the FL protein (Fig S2). Taken together this to me provides very strong evidence that there are no inter-domain interactions in the absence of substrate (as opposed to what the authors write on page 7). I do agree with the conclusion that the protein is a monomer in solution.

Page 8 and 12 and later on in the manuscript: Why are association constants reported for binding events. Please report dissociation constants, as those are independent of the ligand concentration. Also, the units of the association constant is $\text{M}^{-1}\text{s}^{-1}$, not M^{-1} as mentioned in the text.

Fig 2e: please also map the ile residues and please map CSPs on both domains, not only on D1.

Page 11-12: The authors note that 0.1% of the dsRNA is present as ssRNA at the use concentrations and conclude from that small percentage that the fraction of ssRNA can be neglected. I disagree with that. In case the ssRNA binds stronger to the enzyme than the dsRNA does (which is the case) the binding even will remove the ssRNA from the solution after which the dsRNA will partially dissociate. In that manner the small fraction of ssRNA: dsRNA equilibrium will eventually fully shift towards the

ssRNA. Experimentally, the authors also observe that the helicase results in a shift of the dsRNA-ssRNA equilibrium, towards ssRNA (Fig. 4A and S6C). The authors do discuss this in part, but in my eyes this is the only mechanism that is relevant.

Fig 3a: the authors should check if the NMR spectra of the dsRNA titrations are not changing over time. My expectation is that the helicase binds the small fraction of ssRNA that is present in the dsRNA sample. This will then lead to a slow, but continues increase in the concentration of the ssRNA that is bound to the helicase. In other words, I am not convinced that the spectra in Fig 3a (middle and bottom rows) are actually spectra of the helicase bound to dsRNA and would rather think that those are spectra of the helicase bound to ssRNA. The dsRNA does not bind (directly) to the helicase, as is also visible in Fig 3c. The fraction of the (ssRNA) bound helicase likely increases (slowly) over time.

The first paragraph of page 13 contradicts itself. The authors say that dsRNA binding is not compatible with the closed conformation that is adopted with ssRNA (with which I fully agree), but they also say that the dsRNA bound conformation is the same as the ssRNA bound conformation. To me, this can only make sense when the addition of dsRNA results in the formation of a complex between the helicase and ssRNA (as mentioned above).

Fig 4b: the observation that the helicase interactions with RNA decreases when going from UA-12mer to GUCA-12mer to GC-14mer reflects that the DG of duplex formation increases and that thus less ssRNA is (initially) available for the interaction.

Reviewer #2 (Remarks to the Author):

The authors have studied here using primarily NMR spectroscopy the long standing question on how DEAD box protein act as RNA helicase in an ATP dependant manner. They report here convincing evidence that the DEAD-Box helicase acts by binding single-stranded RNA (pushing the equilibrium of available single-stranded RNA from a duplex) although the binding to double-stranded RNA is weak. ATP hydrolysis would be important to release the ssRNA.

The evidence lies on the basis of chemical shift changes in the protein which matches those of the binding of ssRNA, evidence with F-labeled RNA that also matches with the binding of single-stranded RNA and of the fact that weakly stable RNA duplexes are more easily reacting than very stable dsRNA. Although the mechanism per se regarding such helicase activity was shown with DNA, this mechanism was not shown before for this important class of RNA helicase. I do not see major change needed in this solid paper.

As a minor point: I think that a final scheme indicating the role of the ATP hydrolysis for the release of the RNA will add to the paper.

Point-by-point response to the reviewers' comments

We would like to express our sincere gratitude to both referees for their valuable comments and suggestions on our manuscript. Based on the reviewers' insightful comments, we have carefully revised the manuscript. The revisions made to the manuscript are highlighted in red font. We hope that these changes have satisfactorily addressed all of the concerns raised by the reviewers.

Reviewer #1 (Remarks to the Author):

My mayor point is that I disagree with the model that the authors bring up, where the helicase interacts with the dsRNA directly. Rather, the data indicate that the helicase binds to a (small) ssRNA fraction (that is in the dsRNA). This small ssRNA fraction then interacts with the helicase. As a consequence the dsRNA:ssRNA equilibrium shifts further towards the ssRNA side. In case dsRNA would be recognized directly the affinity for any dsRNA would be very similar (the interaction is sequence independent). However, this is something that the authors clearly do not observe. In my opinion there is no (relevant) direct interaction between dsRNA and the helicase. The authors also mention this scenario (page 15: "The 19F NMR results indicate that the binding of DDX3X to RNA greatly stabilizes the ssRNA state, either by binding to the pre-existing ssRNA state and/or by binding to dsRNA to form the ssRNA-DDX3X structure."). The latter option "by binding to dsRNA" does not take place. The dsRNA first (locally or globally) unwinds, after which the helicase interacts with the ssRNA (region). As the data in the paper is in general of high quality my suggestion would be to reinterpret all data and to rephrase the text in the light of a model in which the helicase does not interact with dsRNA at all (which is also in agreement with the literature) and that helicase activity is only due to the interaction between a small fraction of ssRNA which then results in the slow unwinding of the dsRNA due to a shift in the dsRNA-ssRNA equilibrium.

We appreciate the reviewer's critical comment regarding the interpretation of the results. As the reviewer pointed out, many of our results indicate that DDX3X has a stronger affinity for ssRNA compared to dsRNA, which initially led us to propose a model that DDX3X exclusively binds to the pre-existing ssRNA state to shift the ssRNA-dsRNA equilibrium. However, based on the following

observations, we have included the binding to dsRNA to form a locally unwound conformation when remodeling the stable duplex in our final model.

(1) While the binding affinity of DDX3X toward dsRNA is markedly lower than that for ssRNA, weak binding toward dsRNA was still observed in both electrophoretic mobility shift assays (EMSA) and NMR experiments. In the EMSA analyses, the peak height of the band of GC-14mer-FAM was slightly reduced by 10-15% in the presence of 25 μM DDX3X, accompanied by broadening toward the high-molecular-weight direction (Fig. 3c). Similarly, in the ^{19}F NMR experiments observing the 2'- ^{19}F labeled GC-14mer, the ^{19}F signal of the dsRNA state was broadened by adding DDX3X E348Q in the ATP-bound state (Fig. 4b). These results indicate the presence of the transient binding of DDX3X to dsRNA. We think that such weak binding to dsRNA contributes to the formation of the initial encounter complex that precedes the unwinding event. The weak binding toward the stable GC-14mer dsRNA was also observed in the previous study by Epling *et al.* (Epling *et al.*, *J. Mol. Biol.* 427, 1779 – 1796 (2015)), where the band shift of fluorescently labeled GC-14mer was suggested in the presence of $>30 \mu\text{M}$ DDX3X protein. The formation of the dsRNA-DDX3X complex was also reported in the crystal structure solved by Song and Ji (PDB ID: 6O5F, Song and Ji *Nat. Commun.* 10:3085, (2019)), where DDX3X in the nucleotide-free state binds to a 26-bp duplex. These studies support the notion that such a DDX3X-dsRNA complex can be formed as a metastable intermediate state, even though the intrinsic affinity is weak.

(2) The binding of the DDX3X E348Q mutant to the pre-existing ssRNA state does not fully explain the population of the bound state of DDX3X as observed in the HMQC spectra (Fig. 3a). Based on the free energy of the hybridization reaction using the nearest neighbor parameters (-17.8 kcal/mol for GUCA-12mer and -35.0 kcal/mol for GC-14mer), the apparent dissociation constant ($= 1/K_m$, where $K_m = [\text{dsRNA}]/[\text{ssRNA}]^2$) of GUCA-12mer and GC-14mer is calculated to be 2.5×10^{-13} M ($=250$ fM) and 1.6×10^{-25} M at 35 $^\circ\text{C}$, respectively. This corresponds to an ssRNA fraction of 0.035% and $\sim 2.9 \times 10^{-8}$ % at a concentration of 1 μM as a single strand. The absence of free ssRNA was also confirmed through ^{19}F NMR analyses of fluorinated RNAs, where no free ssRNA signal was observed in the ^{19}F spectrum of 2'- ^{19}F labeled GUCA-12mer and GC-14mer (Fig. 4b). If we assume that DDX3X exclusively binds to the pre-existing free ssRNA state, the fraction of the ssRNA-DDX3X complex can be calculated based on the following thermodynamic scheme (Eq. 14 in the main text).

Then, the fractional population of the DDX3X-ssRNA complex can be calculated using the two equilibrium constants, K_m and K_d ($= [DDX3X][ssRNA]/[DDX3X \cdot ssRNA]$). Using the K_d value of 1.7×10^{-5} M ($= 17 \mu\text{M}$) for the DDX3X-ssRNA affinity, we would expect only a small fraction of DDX3X to be bound to ssRNA. For example, the fractional population of the bound state of DDX3X is expected to be as low as 0.046% in the presence of $500 \mu\text{M}$ GUCA-12mer as a single strand. Clearly, this value does not agree with our NMR results where the free and bound populations were comparable when titrating GUCA-12mer (Fig. 3a). This can be understood considering that $1/K_m \gg K_d$, which means the binding affinity of DDX3X to ssRNA does not efficiently compete with the self-hybridization reaction of GUCA-12mer. From these calculations, we interpreted the results that the bound signal represents the locally unwound DDX3X-dsRNA complex. In this complex, DDX3X forms a closed conformation by binding to the ssRNA region of the substrate without fully separating the duplex as schematically illustrated in Fig. 4d bottom. We propose that the formation of this locally unwound conformation represents an important intermediate state that precedes the complete unwinding of the duplex structure.

Figure 4 in the revised manuscript

(d) Cartoon representations of the interaction between each RNA ligand and DDX3X are shown.

(3) The presence of the locally unwound complex is supported by the recently reported crystal structure of DDX3X bound to a DNA-RNA hybrid (PDB: 7LIU, Enemark and Yu, to be published). In this structure, the DDX3X molecule in complex with ADP-BeF₃ (an ATP analog) binds to the single-stranded region at the 5'-terminus of the DNA/RNA hybrid duplex that partially retains a base-paired structure at the 3'-terminal side. Notably, the conformation of DDX3X in the complex is almost perfectly consistent with the closed conformation of the DDX3X/AMPPNP/poly-U₁₀ structural model (RMSD of 0.75Å), which is in line with our model that the formation of the closed conformation of DDX3X underlies the global/local unwinding process of duplexes or any structured RNA elements.

Supplementary Figure 9 in the revised manuscript.

Ribbon diagram of the homology model of DDX3X/AMPPNP in complex with poly-U₁₀ (left) and DDX3X/ADP-BeF₃ in complex with an RNA-DNA hybrid (5'-r(GGGCGGG)d(CCCGCC)-3', where “r” and “d” represent the RNA and DNA nucleotides, respectively) (PDB ID: 7LIU) (center). Only one DDX3X molecule in the unit is shown for clarity. (right) The overlay of the DDX3X structure (cyan/green: DDX3X/ADP-BeF₃/RNA-DNA, pink: DDX3X/AMPPNP/poly-U₁₀). The RMSD value over the pruned set of pairs is shown.

We also note that the relative contribution of each pathway depends on the balance between the affinity of DDX3X for ssRNA and the stability of the duplex, and we expect both pathways can contribute to the unwinding process of physiological RNA substrates. In the case of relatively weakly associating dsRNA like UA-12mer, the binding occurs exclusively through the pre-existing ssRNA state. We have demonstrated that the melting of UA-12mer could be nicely explained by using this model as shown in Fig. 4c in the main text. On the other hand, for very stable duplexes such as GUCA-12mer and GC-14mer, the unwinding reaction would primarily occur through the weak

association with dsRNA to form the locally unwound state as depicted in Fig. 4d bottom. We do not completely rule out the presence of binding between pre-existing ssRNA and DDX3X in the case of very stable duplexes as suggested by the reviewer.

Finally, the reviewer may wonder how this locally unwound DDX3X-dsRNA complex is formed from a microscopic view. As suggested by the reviewer, there are two major pathways for forming the locally unwound DDX3X-dsRNA complex: either the dsRNA locally unwinds first and DDX3 binds to the locally unfolded region, or DDX3X binds to dsRNA first and then locally unfolds the dsRNA. These mechanisms are often referred to as “conformational selection” or “induced fit” mechanisms. As pointed out by Hammes et al. (Hammes et al., *PNAS* 106, 33, 13737–13741 (2009)), it is very challenging to conclusively discriminate between these two mechanisms since the discrimination should be made by comparing the flux of these two pathways, which requires complete knowledge of the populations of all intermediate states and transition rate constants between them. Furthermore, the dominant mechanism strongly depends on the concentration of enzymes and substrates, and quite often the two mechanisms jointly contribute to the formation of the complex. In principle, NMR can provide kinetic information based on quantitative measurements of transition rate constants between free and bound states, as previously done in many other systems (for example, Sekhar *et al. eLife* 7:e32764 (2018)); however, our attempts to measure the binding kinetics were not successful because the on- and off-rates of the DDX3X-RNA complex were too slow to be characterized by ZZ-exchange or CEST methods. Nevertheless, based on our experimental results, we favor the latter induced-fit mechanism, where DDX3X binds to dsRNA first to form a locally unwound structure, as the dominant mechanism. This is because we did not observe any locally unfolded state in the ¹⁹F NMR spectrum of GUCA-12mer and GC-14mer in the absence of DDX3X, indicating that the population of the preformed locally unfolded state is very small if it exists. For the conformational selection pathway to dominate, the association rate constant toward this preformed locally unfolded state needs to be significantly larger than that for the major dsRNA (folded) state by several orders of magnitude. We expect this to be less likely if we assume that the association rate constant is diffusion-limited. Considering that detailed binding kinetics are not available for this system, we have decided not to pursue this topic further in our manuscript, which we believe is beyond the scope of this study.

In the revised manuscript, we have explicitly defined two pathways by which DDX3X binds to dsRNA: the first pathway involves binding to the pre-existing ssRNA, while the second pathway

involves binding directly to dsRNA to form a partially unwound complex. We then quantitatively discuss that the first pathway alone cannot explain the NMR results and emphasize the importance of considering the second pathway for highly stable duplex substrates. Although we did not mention the structure of DDX3X-RNA/DNA hybrid (PDB: 7LIU) in the original manuscript due to the unavailability of experimental details, we have now decided to include it in the discussion as additional support for our model. We would like to thank the reviewer for raising this important point. We believe that these changes have significantly improved the clarity and completeness of our manuscript.

(Page 18 line 4)

Thermodynamic basis of the stabilization of ssRNA by binding to DDX3X

The ^{19}F NMR results indicate that the binding of DDX3X to RNA greatly stabilizes the ssRNA structure, either by binding to the pre-existing ssRNA state or by weakly associating with dsRNA initially and forming a locally unfolded complex accompanied by the deformation of the dsRNA structure. Given that the intrinsic affinity for dsRNA is much weaker than that for ssRNA, the first mechanism is appealing as a general unfolding mechanism of dsRNA. In this scenario, the NMR results can be simply interpreted as a shift in the dsRNA-ssRNA structural equilibrium towards the ssRNA state due to the tight binding of DDX3X to ssRNA, *i.e.* the favorable interaction between DDX3X and ssRNA leads to a predominant population of the ssRNA state within the equilibrium, which might serve as the underlying thermodynamic basis for the unwinding activity of DDX3X.

(Page 19 line 9)

It is interesting to consider whether such simple mass action can also explain the ^{19}F NMR results of GUCA-12mer. The stability of the GUCA-12mer duplex was estimated by using the nearest-neighbor parameters^{49,50}, yielding a ΔG° value of -17.8 kcal/mol. This value can be recast into a K_m value according to the definition of Gibbs free energy ($K_m = e^{-\Delta G^\circ/RT}$) to obtain a K_m value of 4×10^{12} [M⁻¹] at 35 °C ($1/K_m = 250$ [fM]). If we consider a reaction scheme in which two DDX3X molecules globally unwind dsRNA to form two DDX3X-ssRNA complexes ($2[\text{DDX3X}] + [\text{dsRNA}] \rightarrow 2[\text{DDX3X}\cdot\text{ssRNA}]$), the apparent dissociation constant for this process, K'_d , corresponds to $K_m \times K_d^2$ (see Materials and Methods for derivation). **Using the dissociation constant for poly-U₁₀ ($K_d = 17$ [μM]), the K'_d value is estimated to be ~ 1200 [M],**

meaning that the affinity of DDX3X for ssRNA is not strong enough to efficiently compete with the self-hybridization reaction of GUCA-12mer. Under the ^{19}F NMR experimental conditions, where the total concentration of GUCA-12mer is $100\ \mu\text{M}$ as a single strand, the ssRNA population is expected to increase from $0.0035\ \%$ to $0.066\ \%$ by adding $300\ \mu\text{M}$ ATP-bound E348Q DDX3X. However, this increase is much smaller than the observed fractional population of the bound state, which was as high as $\sim 25\%$ in the ^{19}F NMR spectrum (Fig. 4b). The same discussion applies to the NMR experiments observing DDX3X methyl probes. When $500\ \mu\text{M}$ (as a single strand) GUCA-12mer was added to $50\ \mu\text{M}$ ATP-bound E348Q DDX3X, the bound population is calculated to be 0.046% if we only consider the binding of DDX3X to ssRNA, which does not agree with our NMR results where the free and bound populations were comparable (Fig. 3a). These thermodynamic considerations indicate that the results obtained for stable duplexes, such as GUCA-12mer, cannot be fully explained by considering the binding to ssRNA alone. Instead, the alternative pathway involving binding to dsRNA should be included to fully account for the experimental observations. We propose that the signal observed when DDX3X binds to GUCA-12mer does not reflect the global unwinding of the duplex; rather, this signal likely represents a locally unwound state of the duplex upon interaction with DDX3X, as schematically illustrated in Fig. 4d bottom. In this complex, DDX3X forms a closed conformation with ssRNA through the local unfolding of the duplex, while the ligand RNA maintains its base-paired structure without complete separation of the two strands. The formation of such a locally unwound state of dsRNA presumably represents the initial intermediate state preceding the dissociation of dsRNA.

(Page 24 line 1)

From our NMR results in conjunction with thermodynamic considerations of binding affinities, we propose two major pathways for the unwinding of dsRNA. The first pathway involves the binding of DDX3X to pre-existing ssRNA, while the second pathway involves the binding of DDX3X to dsRNA to form a locally unfolded DDX3X-dsRNA complex (Fig. 4d). The relative contribution of each pathway depends on the balance between the affinity of DDX3X for ssRNA and the stability of the duplex, and we expect that both pathways can contribute to the unwinding process of physiological RNA substrates. The first pathway (*Metastable duplex* in Fig. 4d top) predominates when the stability of dsRNA is moderate and the binding affinity of DDX3X for ssRNA (K_d) is comparable to the affinity for the hybridization process (K_m). We demonstrated

that the melting of UA-12mer could be nicely explained by using this model, where the K_d value (=15 [μ M]) was smaller than the $1/K_m$ value (= 28 [μ M]) (Fig. 4c) and the concentration of DDX3X was high enough to shift the equilibrium toward the ssRNA-bound state. For the melting of GUCA-12mer and GC-14mer, on the other hand, the stability of the duplex is much higher than the affinity of DDX3X for ssRNA. Therefore, the second pathway predominates the unwinding process (*Stable duplex* in Fig. 4d bottom). In this pathway, the binding of DDX3X can only partially melt the stable duplex, forming the locally unwound DDX3X-dsRNA complex without completely separating the two strands. This locally unwound state of dsRNA would represent the initial structural intermediate preceding the global unwinding or strand displacement of dsRNA. Intriguingly, the presence of such a complex where DDX3X binds to a locally unwound duplex is supported by the recent crystal structure of DDX3X in complex with an RNA-DNA hybrid (PDB ID: 7LIU, Enemark and Yu, to be published), which was deposited while conducting our research (Supplementary Fig. S9). In this structure, the DDX3X protein bound to ADP-BeF₃ (an ATP analog) interacts with the single-stranded region at the 5' - terminus of one of the strands of the RNA-DNA hybrid duplex, which retains a partially base-paired structure at the 3' -terminal side. Notably, the conformation of DDX3X in the complex is almost perfectly consistent with the closed conformation of the DDX3X/AMPPNP/poly-U₁₀ structural model (RMSD of 0.75 Å), which is in line with our model that the formation of the closed conformation of DDX3X underlies the global/local unwinding process of duplexes or any structured RNA elements (Supplementary Fig. S9). We also note that both scenarios are consistent with the observations that the unwinding activity is inversely correlated with the stability of duplexes^{44,45}, because it is reasonably expected that more free ssRNA is available or the locally unfolded conformation is more easily formed in less stable duplexes.

(Supplementary Figure 9)

Supplementary Figure 9 Structure of DDX3X in complex with remodeled RNA. Ribbon diagram of the homology model of DDX3X/AMPPNP in complex with poly-U₁₀ (left) and DDX3X/ADP-BeF₃ in complex with an RNA-DNA hybrid (5'-r(GGGCGGG)d(CCCGCC)-3', where “r” and “d” represent the RNA and DNA nucleotides, respectively) (PDB ID: 7LIU). Only one DDX3X molecule in the unit is shown for clarity (center). (right) The overlay of the DDX3X structure (cyan/green: DDX3X/ADP-BeF₃/RNA-DNA, pink: DDX3X/AMPPNP/poly-U₁₀). The RMSD value over the pruned set of pairs is shown.

Figure 1b: The temperature should be mentioned in the legend. This is now only done for the 95C control.

We thank the reviewer for the comment. According to the suggestion, we have added the temperature condition (37 °C) in the figure legend as follows.

(Figure 1b)

(b) RNA unwinding assays for DDX3X and its individual domains. The upper band corresponds to the 18mer/36mer dsRNA, while the lower band corresponds to the 18mer ssRNA displaced from dsRNA by the unwinding activity of DDX3X. In the fourth lane labeled with EDTA, 20 mM EDTA was added to the reaction mixture to chelate the cofactor Mg^{2+} as a negative control. The protein concentration was 2 μ M, and the reaction mixture was incubated for 30 mins at 37 °C.

Figure 1c: I am not convinced if the discussion regarding residual domain interactions based on the $S^2_{\text{axis}}\tau_c$ values is conclusive. The τ_c value that is discussed for the FL protein (37 ns) is an overestimation and based on one “outlier” resonance (I213 likely). In case that residue is discarded the estimate of τ_c would drop to around 27 ns, smaller than what is expected from a complex with two domains that form a stable interface (thus proving that the domains are not associating in the apo state). In addition, there are no CSPs between the individual domains and the FL protein (Fig S2). Taken together this to me provides very strong evidence that there are no inter-domain interactions in the absence of substrate (as opposed to what the authors write on page 7). I do agree with the conclusion that the protein is a monomer in solution.

We appreciate the reviewer’s thoughtful comment. As the reviewer pointed out, the $S^2_{\text{axis}}\tau_c$ values of methyl probes, with the exception of Ile213, are generally lower than 37 ns, apparently suggesting the absence of inter-domain interactions in the apo state. It is widely known that $S^2_{\text{axis}}\tau_c$ values

detected on side-chain methyl groups tend to be significantly smaller compared to those measured on main-chain amide groups, mainly due to the intrinsic flexibility of the side chain. Mittermaier *et al.* reported that the order parameters (S^2_{axis}) detected on Ile $\delta 1$ methyl groups typically range between 0.1 and 0.8, while those of Met ϵ methyl groups range between ~ 0 and 0.7 (Mittermaier *et al.*, *JBNMR* 13: 181–185, 1999). Therefore, it is generally challenging to reliably estimate the overall correlation time from the $S^2_{\text{axis}}\tau_c$ value of the side-chain methyl group, and the estimation should be made on the methyl probe that is located inside the core of the protein and whose side-chain motion is highly restricted. In our case, Ile213 fulfills this requirement, and the smaller $S^2_{\text{axis}}\tau_c$ values observed for the rest of the methyl probes likely reflect the intrinsic flexibility of the side chain.

Figure 1. Distribution of order parameters for the backbone amides (S^2_{NH}) and Ala, Thr, Ile, Val, Leu and Met methyl axes (S^2_{axis}) of eight proteins. Values of S^2_{axis} were determined from CH_2D moieties using methods described previously (Muhandiram *et al.*, 1995). Sample sizes are NH: 583, Ala: 46, Thr: 22, Ile $\text{C}^{\gamma 2}$: 30, Val $\text{C}^{\gamma 1,\gamma 2}$: 52, Ile $\text{C}^{\delta 1}$: 33, Leu $\text{C}^{\delta 1,\delta 2}$: 92, Met C^{ϵ} : 17. The proteins studied were apo-calmodulin [3CLN (Babu *et al.*, 1988)], the N-terminal cellulose binding domain from endoglucanase C [1ULO (Johnson *et al.*, 1996)], the C-terminal SH2 domain from phospholipase $\text{C}\gamma 1$ [2PLE (Pascal *et al.*, 1994)], the N-terminal SH3 domain from drk, Ca^{2+} -bound staphylococcal nuclease [1SNC (Loll and Lattman, 1989)], the N-terminal SH2 domain from Syp tyrosine phosphatase [1AYD (Lee *et al.*, 1994)], the N-terminal domain of troponin C [5TNC (Herzberg and James, 1988)] and ubiquitin [1UBQ (Vijay-Kumar *et al.*, 1987)]. Data from flexible N- and C-termini and from the domain linker region of calmodulin were not included in the analysis. In the cases of the C-terminal SH2 domain from phospholipase $\text{C}\gamma 1$ (Kay *et al.*, 1996), the N-terminal SH3 domain from drk (Yang and Kay, 1996), the N-terminal SH2 domain from Syp tyrosine phosphatase (Kay *et al.*, 1998) and the N-terminal domain of troponin C (Gagne *et al.*, 1998), the ^2H S^2_{axis} values have been published (references are listed adjacent to the protein).

Figure 1 from Mittermaier *et al.*, *Journal of Biomolecular NMR* 13: 181–185, 1999

Given the difficulty in estimating the overall τ_c value, our conclusion is primarily based on comparing the full-length protein with the individual domains. As shown in Fig. 1c in the main text, the distribution of τ_c values for the full-length protein (D1-D2) is significantly higher than those of the individual domains, supporting the notion that the full-length protein does not behave like “beads on a string” and that the domain motion is, at least to some extent, restricted.

Figure 1c in the revised manuscript.

(c) Plots of $S^2_{axis} \tau_c$ for Ile and Met methyl groups of DDX3X (D1-D2, black) and its individual domains (D1 and D2, colored blue and green, respectively). Predicted values from HYDROPRO are shown as dotted lines. The measurements were performed at 35 °C. The error bars represent the standard deviations of the fitted rates estimated using the covariance matrix method.

The reason for the very small chemical shift difference between the full-length protein and the individual domains is simply because methyl probes are not available at the domain interface. In addition, the domain interface between the D1 and D2 domains is relatively small ($\sim 710 \text{ \AA}^2$) and only 5-6 residues are involved in the inter-domain interaction in the crystal structure (PDB ID: 5E7I), suggesting that the structural perturbation caused by the interdomain interaction can be quite small.

In response to the reviewer's comment, however, we conducted an additional NMR experiment to directly probe the presence of inter-domain interactions (see Supplementary Figure S3 below). We introduced a methyl probe, M531, by mutating R531, and measured a ^{13}C -edited NOESY spectrum focusing on interdomain methyl-methyl contacts. In the structural model of the R531M variant, the side chain of M531 on the D2 domain is in close proximity to M355 on the D1 domain ($\text{C}\epsilon\text{-C}\epsilon$ distance of $\sim 5.3 \text{ \AA}$), thus the presence of the inter-domain interaction can be directly monitored by observing an NOE cross-peak between the M531 and M355 methyl resonances. In the NOESY spectrum recorded with a mixing time of 500 ms, the cross-peak between these two Met resonances was not observed, indicating that the inter-domain interaction between the D1 and D2 domains is weak and that there is considerable flexibility at the D1-D2 domain interface. We also confirmed that a cross-peak originating from the intradomain interaction between M531 and I507 ($\text{C}\epsilon\text{-C}\delta 1$ distance of $\sim 4.7 \text{ \AA}$) was clearly observed in the NOESY spectrum, and the overall $S^2_{axis} \tau_c$ distribution was not significantly perturbed in the R531M variant. Taken together with the above interpretation of the

overall $S^2_{\text{axis}}\tau_c$ distribution, we have now revised our interpretation of the interdomain interaction. We now conclude that there is significant flexibility at the domain interface, while the major orientation of the two domains remains consistent with the crystal structure. Indeed, the presence of such inter-domain flexibility might play a role in the recognition of the ssRNA substrate.

We have included the above inter-domain NOE experiments and rewrote the results and discussions regarding the inter-domain flexibility. This reinterpretation does not alter the main conclusion of our paper.

(Page 7 line 14)

The overall τ_c value was estimated to be ~ 37 ns based on the highest $S^2_{\text{axis}}\tau_c$ value (for I213) assuming isotropic rotational tumbling. This value was in reasonable agreement with the calculated value of 33.5 ns from HYDROPRO^{35,36} using the crystal structure of the adenosine monophosphate (AMP)-bound form of DDX3X (PDB ID: 5E7J)¹⁹. **The $S^2_{\text{axis}}\tau_c$ values for the rest of the methyl probes were generally lower than this value, reflecting the side-chain flexibility of Ile and Met side chains whose S^2_{axis} value is typically distributed around 0.1 to 0.8 and ~ 0 to 0.7, respectively³⁷. This overall τ_c estimation suggests that the rotational tumbling of the D1 and D2 domains is to some extent restricted in the full-length DDX3X context, likely due to the formation of transient interdomain interactions even in the apo state. Additionally, these results support the notion that DDX3X predominantly exists as a monomer under our experimental conditions.**

While the overall τ_c estimation is consistent with the HYDROPRO estimation based on the crystal structure, the very small chemical shift difference between the tandem D1-D2 and the individual domains (Supplementary Figs. S2a and S2b) strongly suggests that the interaction between these two domains is rather weak and that there is structural flexibility at the D1-D2 domain interface. To more directly characterize this inter-domain interaction, we introduced a methyl probe, M531, at the D1-D2 domain interface by replacing R531, and then measured an HMQC-nuclear Overhauser spectroscopy (NOESY) spectrum focusing on methyl-methyl contacts between these two domains. In the modeled structure of the R531M variant based on the apo DDX3X crystal structure (PDB ID: 5E7I)¹⁹, the side chain of M531 on the D2 domain is close to the M355 side chain on the D1 domain (C ϵ -C ϵ distance of ~ 5.3 Å) (Supplementary Fig. S3a), which allows us to monitor the presence of the inter-domain interaction directly

through an NOE cross-peak between the M531 and M355 methyl resonances. The Met methyl signal of M531 could be readily assigned by comparing the spectra of the wild type and the R531M variant (Supplementary Fig. S3b). In the NOESY spectrum recorded with a mixing time of 500 ms, the cross-peak between these two Met resonances was not observed, indicating the presence of structural flexibility and/or heterogeneity at the D1-D2 domain interface (Supplementary Fig. S3c). We also confirmed that a cross-peak originating from the intra-domain interaction between M531 and I507 (C_{ϵ} - $C_{\delta 1}$ distance of ~ 4.7 Å) was clearly observed in the NOESY spectrum, and that the overall $S^2_{axis}\tau_c$ distribution was not significantly affected in the R531M variant (Supplementary Fig. S3d). Taken together with the interpretation of the overall τ_c value, these results indicate the presence of structural flexibility at the D1-D2 interdomain interface while retaining the major domain configuration consistent with the crystal structure. Such domain flexibility likely plays a role in the recognition of the RNA substrate, as will be described in detail below.

(Page 22 line 10)

Several studies have proposed the functional cycle for dsRNA unwinding, involving the apo DDX, the pre-unwound DDX-dsRNA complex, and the post-unwound DDX-ssRNA complex, where the domain reorganization induced by the binding and hydrolysis of ATP facilitates the unwinding of dsRNA^{18,20,21}. Our NMR results demonstrated that, although there is some structural flexibility and/or heterogeneity at the D1-D2 domain interface, the domain motion is rather restricted, with the major conformation consistent with the crystal structure with the closed D1-D2 interface. Importantly, we found that the interdomain dynamics are not strongly coupled to the AMPPNP/ADP status, indicating that domain reorganization is not tightly linked to the hydrolysis of ATP during the RNA unwinding reaction.

(Page 23 line 14)

Thus, these favorable interactions serve as the thermodynamic basis for the formation of the DDX-ssRNA complex. We also note that the structural heterogeneity/flexibility at the D1-D2 domain interface would facilitate the rearrangement of the D1 and D2 domains to form this closed structure.

(Supplementary Figure S3)

Supplementary Figure 3 Interdomain interaction of DDX3X. (a) Close-up view of the interface formed between the D1 and D2 domains of DDX3X in the apo state (PDB ID: 5E7I). R531 was replaced with Met by using the ChimeraX *swapa* module. The C_ϵ - C_ϵ distance between M355 and M531 and the C_ϵ - $\text{C}_\delta 1$ distance between M531 and I507 are displayed. (b) Overlay of the ^{13}C - ^1H HMQC spectra of [$\text{u}-^2\text{H}$; Ile $\delta 1$ - $^{13}\text{C}^1\text{H}_3$; Met ϵ - $^{13}\text{C}^1\text{H}_3$]-labeled wild-type (coral, single contour) and [$\text{U}-^2\text{H}$; Ile $\delta 1$ - $^{13}\text{C}^1\text{H}_3$; Met $\alpha\beta\gamma$ - ^2H , ϵ - $^{13}\text{C}^1\text{H}_3$]-labeled R531M variant (navy, multiple contours) DDX3X. The spectrum of the wild type was shifted by 0.1 ppm in the ^{13}C dimension to correct for the isotope shift. (c) HMQC-($^{13}\text{C}-t_1$)-NOESY ($^1\text{H}-t_2$) (500 ms mixing time) spectrum of the R531M variant DDX3X. (d) Correlation plot of the $S^2_{\text{axis}\tau_c}$ values of wild-type and R531M variant DDX3X. The error bars represent the standard deviations of the fitted rates estimated using the covariance matrix method. r^2 is Pearson's correlation coefficient squared.

Page 8 and 12 and later on in the manuscript: Why are association constants reported for binding events. Please report dissociation constants, as those are independent of the ligand concentration. Also, the units of the association constant is $M^{-1}s^{-1}$, not M^{-1} as mentioned in the text.

We appreciate the reviewer's suggestion and apologize for the confusion. We reported the binding affinity using K_a (unit of M^{-1}), which is the inverse of the dissociation constant K_d (unit of M), not the second-order association rate constant (unit of $M^{-1}s^{-1}$). To avoid confusion, we rewrote all of the binding affinity values using K_d . For representing the stability of the duplex, however, we decided to continue using the association constant (K_m) which has a unit of M^{-1} to be consistent with previous studies. In this representation, a larger K_m (more negative ΔG) means a more stable duplex. When comparing the affinity of duplexes with the binding affinity of DDX3X-RNA, we have included the value of $1/K_m$ (unit of M), which is equivalent to K_d (unit of M), to guide the readers.

(Page 9 line 11)

The **dissociation constants** of AMPPNP and ADP were estimated to be $910 \pm 150 \mu M$ and $51 \pm 7.9 \mu M$, respectively, from two-dimensional NMR line-shape analyses³⁸ (Supplementary Fig. S4b).

(Page 13 line 18)

We measured ^{13}C - 1H HMQC spectra of ATP-bound DDX3X E348Q in the presence of varying concentrations of GUCA-12mer and GC-14mer dsRNA (Fig. 3a). As a control, we first conducted a titration experiment using poly- U_{10} ssRNA and obtained an apparent **dissociation constant of $17 \pm 1.8 [\mu M]$** by fitting the intensity ratio of signals from the poly- U_{10} free (F) and bound (B)/closed state (Fig. 3b).

(Page 18 line 15)

As proof of this concept, we quantitatively analyzed the UA-12mer titration profiles, considering that DDX3X binds exclusively to ssRNA for simplicity. The binding process can then be described by assuming a 3-state thermodynamic model, comprising the unbound dsRNA state,

the unbound ssRNA state, and the DDX3X-bound ssRNA state. Each of these states corresponds to the three distinct signals observed in the ^{19}F NMR spectra. These three states are related by the two equilibrium constants: K_m ($= [\text{dsRNA}]/[\text{ssRNA}]^2$) and K_d ($=([\text{DDX3X}][\text{ssRNA}])/[\text{DDX3X}\cdot\text{ssRNA}]$), which describe the hybridization of UA-12mer and the association between ssRNA and DDX3X, respectively (see Materials and Methods for details). The fractional populations of each state were estimated from the signal intensities of the three states, and then the populations were fit to the above model to obtain the two equilibrium constants. The titration profiles could be well fit to the model yielding the best-fit values: $K_m = 36,000 \pm 4,700 [\text{M}^{-1}]$ ($1/K_m = 28 \pm 4.0 [\mu\text{M}]$) and $K_d = 15 \pm 1.9 [\mu\text{M}]$ (Fig. 4c). Notably, the obtained K_d value was in good agreement with the dissociation constant for poly-U₁₀ ($= 17 \pm 1.8 [\mu\text{M}]$) (Fig. 3b), confirming that the preferential binding to ssRNA can solely explain the titration profile of UA-12mer (Fig. 4d, top).

It is interesting to consider whether such simple mass action can also explain the ^{19}F NMR results of GUCA-12mer. The stability of the GUCA-12mer duplex was estimated by using the nearest-neighbor parameters^{49,50}, yielding a ΔG° value of -17.8 kcal/mol. This value can be recast into a K_m value according to the definition of Gibbs free energy ($K_m = e^{-\Delta G^\circ/RT}$) to obtain a K_m value of $4 \times 10^{12} [\text{M}^{-1}]$ at 35 °C ($1/K_m = 250 [\text{fM}]$). If we consider a reaction scheme in which two DDX3X molecules globally unwind dsRNA to form two DDX3X-ssRNA complexes ($2[\text{DDX3X}] + [\text{dsRNA}] \rightarrow 2[\text{DDX3X}\cdot\text{ssRNA}]$), the apparent dissociation constant for this process, K'_d , corresponds to $K_m \times K_d^2$ (see Materials and Methods for derivation).

(Page 36 line 10)

The binding of poly-U₁₀ ssRNA to DDX3X was analyzed by assuming a simple one-site binding model as follows:

where $[\text{DDX3X}]$, $[\text{ssRNA}]$, and $[\text{DDX3X}\cdot\text{ssRNA}]$ denote the molar concentrations of DDX3X in the free state, unbound ssRNA, and DDX3X bound to ssRNA, respectively. The dissociation constant for the binding of ssRNA to DDX3X is given by:

$$K_d = \frac{[\text{DDX3X}][\text{ssRNA}]}{[\text{DDX3X} \cdot \text{ssRNA}]} \quad [11]$$

and the total protein concentration, C_T , and total ligand ssRNA concentration, L_T , are given by:

$$C_T = [DDX3X] + [DDX3X \cdot ssRNA] \quad [12]$$

$$L_T = [ssRNA] + [DDX3X \cdot ssRNA]$$

Then, each concentration term can be readily calculated by using K_d , C_T , and L_T as follows:

$$[ssRNA] = \frac{-C_T + L_T - K_d + \sqrt{(C_T - L_T + K_d)^2 + 4K_d L_T}}{2} \quad [13]$$

$$[DDX3X] = C_T - L_T + [ssRNA]$$

$$[DDX3X \cdot ssRNA] = L_T - [ssRNA]$$

(Page 39 line 13)

Assuming that the population of the intermediate state is negligibly small, the apparent dissociation constant for this process, K'_d , is related to K_m and K_d as follows:

$$K'_d = \frac{[DDX3X]^2 [dsRNA]}{[DDX3X \cdot ssRNA]^2} = \frac{[dsRNA]}{[ssRNA]^2} \cdot \left(\frac{[DDX3X][ssRNA]}{[DDX3X \cdot ssRNA]} \right)^2 = K_m K_d^2 \quad [18]$$

(There are a few other similar changes not highlighted above.)

Fig 2e: please also map the ile residues and please map CSPs on both domains, not only on D1.

We appreciate the reviewer's suggestion. Since the complete assignments of Ile methyl probes in the ssRNA-bound/closed conformation were not available due to severe signal overlaps and broadenings, we were not able to calculate the CSP values for Ile probes and did not include Ile probes in the mapping in our original manuscript. For completeness, we have conducted semi-quantitative CSP analyses on the Ile methyl probes as summarized in Fig. S5a in the supporting information. Large CSPs and/or significant signal broadenings were observed for Ile methyl probes located near the ssRNA and AMPPNP/ATP binding sites in both D1 and D2 domains, which is consistent with the proposed model structure of the DDX3X-ssRNA complex.

There is only one Met methyl probe in the D2 domain, Met574, which is located in the C-terminal tail region whose density was not observed in the VASA-polyU₁₀ complex. We have included M574

in the mapping in Fig. 2e using a schematic representation. The chemical shift of M574 did not change upon binding to poly-U₁₀ ssRNA, which is consistent with the structural model of the DDX3X-polyU₁₀ complex.

To clarify these points, we have revised the figures and added explanations in the main text.

(Page 11 line 14)

The chemical shift change upon the formation of the closed conformation was observed in virtually all Ile and Met methyl probes, consistent with the large conformational rearrangements upon binding to poly-U₁₀ as observed in the crystal structural analyses (Figs. 2d and 2e, and Supplementary Figs. S5a and S5b). We observed a marked chemical shift difference in the M221 and M380 methyl probes, which likely reflects the structural rearrangement of a hydrophobic cluster in the D1 domain linking the AMPPNP/ATP-binding cleft to the poly-U₁₀ binding site (Supplementary Fig. S5c). **Since the complete set of assignments for Ile methyl probes in the poly-U₁₀ bound state was not available due to severe signal overlaps and broadenings, we analyzed the chemical shift perturbation of Ile methyl probes by examining the disappearance of the free state signals (Supplementary Figs. S5a and S5b). The marked chemical shift changes and/or signal broadenings were observed in Ile methyl probes located in both the D1 domain (I158, I166, I190, I191, I195, I214, I268, I336, I364, I389, I401) and the D2 domain (I415 and I529), while those distant from the poly-U₁₀ binding site did not exhibit a chemical shift change (I514 and I550). The chemical shift difference was observed not only at the poly-U₁₀ binding interface but also at the AMPPNP/ATP binding site, consistent with the presence of allosteric coupling between these two sites^{11,44}. In addition, the poly-U₁₀ binding pocket within the closed structure of DDX3X, which was modeled from the VASA/AMPPNP/poly-U₁₀ complex¹⁵, showed a positive electrostatic surface that complements the negative electrostatic charge of the poly-U₁₀ RNA (Supplementary Fig. S5d).**

(Figure 2d, 2e)

(d) Overlay the ¹³C-¹H HMQC spectra of [Frac-²H; Ileδ1-¹³C¹H₃; Metε-¹³C¹H₃]-labeled DDX3X E348Q recorded with (orange-red) and without (navy) poly-U₁₀. (e) Mapping of methionine residues that showed significant chemical shift changes upon the binding to poly-U₁₀ onto the modeled structure of DDX3X/AMPPNP/poly-U₁₀. Methionine methyl carbons are shown as spheres. The Met methyl probes that showed significant chemical shift perturbation (CSP) are colored orange, while those with small or undefined CSP are colored gray. M574 is located in the C-terminal tail region whose structure was not modeled from the VASA/AMPPNP/poly-U₁₀ crystal structure (PDB ID: 2DB3).

(Supplementary Figure S5a. b)

Supplementary Figure 5 Interaction of the E348Q variant DDX3X with poly-U₁₀. (a) Ile

and Met methyl region of the ^{13}C - ^1H HMQC spectra of the E348Q variant DDX3X with (orange-red) and without (navy) 2 equimolar poly-U₁₀. The signal at ^1H and ^{13}C chemical shift of 1.28 ppm and 11.0 ppm is from triethylamine (TEA), a counter-ion of the poly-U₁₀ RNA. The assignments of the Met methyl signal that showed a significant chemical shift difference are shown. NMR measurements were performed at 35 °C and 600 MHz in the presence of 5 mM ATP/MgCl₂. (b) Mapping of Met (top) and Ile (bottom) residues that showed significant chemical shift changes upon binding to poly-U₁₀ onto the modeled structure of DDX3X/AMPPNP/poly-U₁₀. Met C ϵ and Ile C δ 1 carbons are shown as spheres. The Met methyl probes that showed significant chemical shift perturbation (CSP) are colored orange, while those with small or undefined CSP are colored gray. M574 is located in the C-terminal tail region whose structure was not modeled from the VASA/AMPPNP/poly-U₁₀ crystal structure (PDB ID: 2DB3). Since the complete set of assignments for Ile methyl probes in the poly-U₁₀ bound state was not available due to severe signal overlaps and broadenings, we analyzed the chemical shift perturbation of Ile methyl probes by examining the disappearance of the free state signals. The Ile methyl probes with large CSP and/or marked intensity reduction are colored orange, while those with small CSP are colored blue. The methyl probes with undefined CSP are colored gray.

Page 11-12: The authors note that 0.1% of the dsRNA is present as ssRNA at the use concentrations and conclude from that small percentage that the fraction of ssRNA can be neglected. I disagree with that. In case the ssRNA binds stronger to the enzyme than the dsRNA does (which is the case) the binding even will remove the ssRNA from the solution after which the dsRNA will partially dissociate. In that manner the small fraction of ssRNA: dsRNA equilibrium will eventually fully shift towards the ssRNA. Experimentally, the authors also observe that the helicase results in a shift of the dsRNA-ssRNA equilibrium, towards ssRNA (Fig. 4A and S6C). The authors do discuss this in part, but in my eyes this is the only mechanism that is relevant.

We thank the reviewer for the comment. We agree that the binding of DDX3X to ssRNA can affect the dsRNA-ssRNA equilibrium, and therefore the presence of a small amount of ssRNA cannot always be ignored. We have rephrased the sentence to acknowledge the potential contribution of the

free ssRNA fraction.

(Page 13 line 13)

GC-14mer shows exceptionally high stability due to its 100% GC content and has been used as a model dsRNA ligand in previous studies^{11,18}. We note that the fraction of ssRNA is estimated to be below ~0.1% at micromolar concentrations. Thus, it is reasonably expected that the spectral changes upon the addition of these RNA molecules primarily reflect the binding of DDX3X to the dsRNA state.

As we mentioned in our response to the major point, the degree to which the binding of DDX3X to ssRNA shifts the dsRNA-ssRNA equilibrium depends on both the stability of the duplex and the binding affinity of DDX3X toward ssRNA. Since the stability of GC-14mer and GUCA-12mer is very high, the binding of DDX3X to pre-existing ssRNA does not effectively shift the dsRNA-ssRNA equilibrium. For example, in the case of GUCA-12mer, the fraction of ssRNA state ($[ssRNA] + [DDX3X-ssRNA]$) increases from 0.0035 % to 0.066 % upon the addition of 300 μ M DDX3X assuming a total RNA concentration of 100 μ M as a single strand. In the ¹⁹F NMR experiments observing 2'-¹⁹F GUCA-12mer, however, we clearly observed an increase in the fractional population of the DDX3X-bound state up to ~25% (Fig. 4b). These NMR results indicate that the binding of DDX3X to the pre-existing ssRNA state alone does not fully explain the experimental observations, and that DDX3X can weakly associate with dsRNA to form a locally unwound DDX3X-dsRNA complex.

In the revised manuscript, we have included the above thermodynamic considerations. We have more explicitly defined the presence of two pathways and emphasized that both of these two pathways contribute to the unwinding process. Then, we discuss that the binding to pre-existing ssRNA alone cannot fully explain the NMR results obtained from highly stable duplex substrates (Please refer to our reply to the major point).

Fig 3a: the authors should check if the NMR spectra of the dsRNA titrations are not changing over time. My expectation is that the helicase binds the small fraction of ssRNA that is present in the dsRNA sample. This will then lead to a slow, but continues increase in the

concentration of the ssRNA that is bound to the helicase. In other words, I am not convinced that the spectra in Fig 3a (middle and bottom rows) are actually spectra of the helicase bound to dsRNA and would rather think that those are spectra of the helicase bound to ssRNA. The dsRNA does not bind (directly) to the helicase, as is also visible in Fig 3c. The fraction of the (ssRNA) bound helicase likely increases (slowly) over time.

We appreciate the reviewer's suggestion. In order to confirm that the NMR sample is in equilibrium, we recorded HMQC spectra of ATP-bound E348Q DDX3X in the presence of 6 eq. GC-14mer (300 μ M) every 3 hours and monitored the signal intensity of the ssRNA-bound signals up to 12 hours. We did not observe an increase in the intensity of the bound state over time, indicating that the NMR sample was already in an equilibrium state at the beginning of the experiment. We did not trace the signal change further longer due to the slow hydrolysis of ATP catalyzed by DDX3X (note that the ATPase activity is not completely abolished in the E348Q mutant).

In response to this suggestion, we have included the time-course experiment in the manuscript as follows.

(Page 14 line 4)

When titrating with poly-U₁₀ ssRNA, the bound percentage exceeded ~70 % upon the addition of 2 equimolar (eq.) amounts of poly-U₁₀, while the bound percentage remained below 40 % even with 10 eq. amounts (as a single strand) of GC-14mer. **We confirmed that the intensity of the bound state signals did not change over a period of 12 hours, indicating that the NMR sample had already reached an equilibrium condition (Supplementary Fig. S6).**

(Supplementary Figure 6)

Supplementary Figure 6 Equilibrium condition of the DDX3X-GC-14mer dsRNA interaction. ¹³C-¹H HMQC spectra of [Frac-²H; Ile δ 1-¹³C¹H₃; Met ϵ -¹³C¹H₃]-labeled DDX3X E348Q (50 μM) in the presence of 6 eq. (300 μM) GC-14mer dsRNA were recorded every 3 hours. The 1D slices of the bound state signal of M221 and M370 are shown in light blue. The 1D projections of the dotted region containing the free and bound signals for M330 are shown in each spectrum. Duplex stability (ΔG°) at 35 °C was estimated by using nearest-neighbor parameters in 1 M NaCl. All NMR measurements were performed at 35 °C and 600 MHz.

The first paragraph of page 13 contradicts itself. The authors say that dsRNA binding is not compatible with the closed conformation that is adopted with ssRNA (with which I fully agree), but they also say that the dsRNA bound conformation is the same as the ssRNA bound conformation. To me, this can only make sense when the addition of dsRNA results in the formation of a complex between the helicase and ssRNA (as mentioned above).

We appreciate the reviewer's critical comment. We agree that the first paragraph on page 13 was misleading. The closed conformation can be interpreted in two ways: as the formation of a DDX3X-ssRNA complex through binding to the pre-existing ssRNA state, or as the formation of a DDX3X-dsRNA complex accompanied by significant deformation of the dsRNA structure (*i.e.*, DDX3X binds

to a locally unfolded ssRNA region within the dsRNA ligand). In the latter complex, the DDX3X-bound dsRNA consists of both dsRNA and ssRNA portions, as schematically depicted in Fig. 4d. In this complex, the binding site of DDX3X on RNA is locally unfolded to form an ssRNA structure. Therefore, the closed conformation of DDX3X can be formed without any steric clash. We would like to emphasize that the latter possibility better explains our NMR results, as we mentioned in our response to the major point.

To clarify this point, we have rephrased the first paragraph on page 13 (page 14 in the revised manuscript).

(Page 14 line 21)

The NMR results obtained so far indicate that, in the presence of an excess amount of dsRNA, DDX3X adopts a closed conformation similar to what has been observed in its complex with ssRNA by weakly associating with the dsRNA substrate. As pointed out previously¹⁵, this closed conformation is not compatible with dsRNA binding because one of the dsRNA strands sterically clashes with the helix in the D1 domain (residues 357 to 366, often referred to as the “wedge helix”) when the dsRNA structure is aligned to the bound ssRNA (Fig. 3d). Therefore, the observed closed conformation should be interpreted either as the formation of a DDX3X-ssRNA complex through binding to the pre-existing ssRNA state, or as the formation of a DDX3X-dsRNA complex accompanied by significant deformation of the dsRNA structure (*i.e.*, DDX3X binds to a locally unfolded ssRNA region within the dsRNA ligand). To obtain further structural insights into the interaction between DDX3X and dsRNA, we then turned to the direct NMR observation of dsRNA ligands and conducted detailed thermodynamic analyses of the complex formation.

Fig 4b: the observation that the helicase interactions with RNA decreases when going from UA-12mer to GUCA-12mer to GC-14mer reflects that the DG of duplex formation increases and that thus less ssRNA is (initially) available for the interaction.

We appreciate the reviewer’s comment, with which we agree. In the pathway where DDX3X binds

to the pre-existing ssRNA to unwind the duplex, the decrease in duplex stability results in an increase in the ssRNA fraction. This, in turn, leads to an increase in the population of the DDX3X-ssRNA state. We clearly observed this trend in the DDX3X titration experiment observing the 2'-fluorinated UA-12mer. We expect that a similar mechanism also applies to the other pathway, where DDX3X binds to dsRNA to form a locally unwound structure, because it is reasonably expected that the locally unfolded conformation is more easily formed in less stable duplexes. In this regard, the increase in the DDX3X-dsRNA interaction in less stable duplexes is consistent with both mechanisms.

To clarify this point, we have included an explanation of the correlation between duplex stability and unwinding activity as follows.

(Page 25 line 7)

We also note that both scenarios are consistent with the observations that the unwinding activity is inversely correlated with the stability of duplexes^{44,45}, because it is reasonably expected that more free ssRNA is available or the locally unfolded conformation is more easily formed in less stable duplexes.

4. Reviewer #2 (Remarks to the Author):

The authors have studied here using primarily NMR spectroscopy the long standing question on how DEAD box protein act as RNA helicase in an ATP dependant manner. They report here convincing evidence that the DEAD-Box helicase acts by binding single-stranded RNA (pushing the equilibrium of available single-stranded RNA from a duplex) although the binding to double-stranded RNA is weak. ATP hydrolysis would be important to release the ssRNA.

The evidence lies on the basis of chemical shift changes in the protein which matches those of the binding of ssRNA, evidence with F-labeled RNA that also matches with the binding of single-stranded RNA and of the fact that weakly stable RNA duplexes are more easily reacting than very stable dsRNA. Although the mechanism per se regarding such helicase activity was shown with DNA, this mechanism was not shown before for this important class of RNA helicase. I do not see major change needed in this solid paper.

As a minor point: I think that a final scheme indicating the role of the ATP hydrolysis for the

release of the RNA will add to the paper.

We greatly appreciate the reviewer's positive comment on our manuscript and thoughtful suggestion. According to the suggestion, we have included the role of ATP hydrolysis in the scheme in Fig. 4d and added a reference to this panel when explaining the role of ATP hydrolysis that is followed by the release of the RNA substrate.

(Page 23 line 17)

Although we mainly used the ATPase-deficient E348Q variant of DDX3X to stabilize the DDX-ssRNA complex, it is important to note that the bound ATP is rapidly hydrolyzed in the wild-type DDX3X and the bound RNA is subsequently released in the ADP-bound state (Fig. 4d). This is also in line with the previous findings that the hydrolysis of ATP contributes to the turnover of the reaction, while it is not necessary for the unwinding activity itself⁴⁰⁻⁴³.

(Figure 4)

(d) Cartoon representations of the interaction between each RNA ligand and DDX3X are shown.

REVIEWERS' COMMENTS

Reviewer #1 (Remarks to the Author):

The authors have very impressively rebutted all my remarks and criticism. The modifications to the text and figures and the additional data are highly convincing. I can now recommend publication of this impressive manuscript without any additional changes. I congratulate the authors with this impressive work.

Point-by-point response to the reviewers' comments

Reviewer #1 (Remarks to the Author):

The authors have very impressively rebutted all my remarks and criticism. The modifications to the text and figures and the additional data are highly convincing. I can now recommend publication of this impressive manuscript without any additional changes. I congratulate the authors with this impressive work.

We would like to express our sincere gratitude to the reviewer for carefully reviewing our manuscript and providing us with many insightful comments, which have significantly improved the manuscript.